# A neural circuit mechanism for regulating vocal variability during song learning in zebra finches

Jonathan Garst-Orozco[1,2], Baktash Babadi[1,3], Bence P Ölveczky[1,4]*

[1]Center for Brain Science, Harvard University, Cambridge, United States; [2]Program in Neuroscience, Harvard University, Cambridge, United States; [3]Swartz Program in Theoretical Neuroscience, Harvard University, Cambridge, United States; [4]Department of Organismic and Evolutionary Biology, Harvard University, Cambridge, United States

**Abstract** Motor skill learning is characterized by improved performance and reduced motor variability. The neural mechanisms that couple skill level and variability, however, are not known. The zebra finch, a songbird, presents a unique opportunity to address this question because production of learned song and induction of vocal variability are instantiated in distinct circuits that converge on a motor cortex analogue controlling vocal output. To probe the interplay between learning and variability, we made intracellular recordings from neurons in this area, characterizing how their inputs from the functionally distinct pathways change throughout song development. We found that inputs that drive stereotyped song-patterns are strengthened and pruned, while inputs that induce variability remain unchanged. A simple network model showed that strengthening and pruning of action-specific connections reduces the sensitivity of motor control circuits to variable input and neural 'noise'. This identifies a simple and general mechanism for learning-related regulation of motor variability.

*For correspondence: olveczky@ fas.harvard.edu

**Competing interests:** The authors declare that no competing interests exist.

## Introduction

Our capacity to learn and reliably execute motor skills underlies much of what we do. Motor skill learning is characterized by high initial motor variability that is gradually reduced as performance improves and skills are consolidated (*Figure 1A*) (*Lee et al., 1999*; *Park et al., 2013*). Variability in motor output can be beneficial early in learning as it allows the motor system to explore a range of actions and selectively reinforce ones that improve performance (*Sutton and Barto, 1998*; *Tumer and Brainard, 2007*; *Wu et al., 2014*). But as viable solutions are found (a good tennis serve, for example), variability in motor output can become detrimental for expert performance and is often reduced. This capacity of the nervous system to regulate variability and plasticity in motor output as a function of learning or skill level enables new skills to be acquired and those already mastered to be stably expressed and maintained. However, the neural circuit mechanisms that regulate motor variability as a function of learning have not been identified.

Addressing this question means linking learning-related changes in the motor circuits underlying skilled performance to a reduction in the variability of their action-related dynamics. While it is known that motor practice induces structural and functional changes in motor control circuits (*Rioult-Pedotti et al., 1998*; *Xu et al., 2009*; *Wang et al., 2011*; *Fu et al., 2012*), it is not clear how such modifications influence overall network function or how they lead to reduced neural and behavioral variability (*Peters et al., 2014*). Moreover, studies of plasticity in motor circuits have tended to focus on connectivity within anatomically confined motor regions (*Sanes and Donoghue, 2000*; *Adkins et al., 2006*), leaving open

**eLife digest** 'Practice makes perfect' captures the essence of how we learn new skills. When learning to play a musical instrument, for example, it often takes hours of practice before we can play a single piece of music properly for the first time. And as we get better, the variability in our performance—which is an advantage during the early stages of learning—becomes less. Likewise, songbirds need lots of practice in order to master the intricate songs they need to sing to attract mates.

Studies in songbirds show that the neural circuits in the brain that are responsible for producing song and for generating vocal variability both converge on a motor control region called the robust nucleus of the arcopallium (or RA for short). However, the details of how learning a song leads to reduced variability in vocal performance are poorly understood.

Now Garst-Orozco et al. have investigated the relationship between learning and variability by studying brain slices of zebra finches. Their experiments reveal that the inputs received by RA neurons from a higher-order brain region that controls song change with practice, with some inputs becoming stronger and others being eliminated as the birds' singing ability improves. However, inputs received by RA neurons from the circuit that generates vocal variability do not change despite the song becoming increasingly precise.

Using a computer simulation, Garst-Orozco et al. show that the sensitivity of RA neurons to variable or 'noisy' input is reduced when inputs from the brain region that controls song are adaptively strengthened and eliminated. This ensures that when the notes and syllables that make up the bird's song have finally been learned, they will be uttered with high fidelity and precision. Intriguingly, motor skill learning in mammals have been associated with neural connectivity changes very similar to those described by Garst-Orozco et al., suggesting that insights from songbirds may lead to a better understanding of how 'practice makes perfect' also works in humans.

the question of how motor learning modifies connections between functionally distinct motor areas. Consequently, the logic by which motor skills are acquired and consolidated in neural circuitry and how this may contribute to reduced variability in motor output has remained elusive.

The zebra finch, a songbird, presents an experimentally tractable model in which to address these questions (*Brainard and Doupe, 2002*; *Mooney, 2009*; *Ölveczky and Gardner, 2011*). Male birds learn a complex motor sequence—their courtship song—by first memorizing the song of a tutor, then engaging in trial-and-error learning to match their vocalizations to a stored template of their tutor's song (*Immelmann, 1969*; *Tchernichovski et al., 2001*). As with human motor skill learning, birdsong learning is characterized by gradual improvements in performance (similarity to the tutor song) and decreased motor variability (*Tchernichovski et al., 2001*; *Ölveczky et al., 2011*) (*Figure 1A,B*).

Two main pathways are involved in the acquisition and production of the bird's song. The *Vocal Motor Pathway (VMP),* which comprises HVC (used as proper name) and the *robust nucleus* of the arcopallium (RA) (*Figure 1C,D*), encodes and controls learned vocalizations (*Simpson and Vicario, 1990*; *Yu and Margoliash, 1996*; *Leonardo and Fee, 2005*). The *Anterior Forebrain Pathway* (AFP), a song-specialized basal-ganglia-thalamo-cortical circuit, provides input to the VMP at the level of RA and is necessary for vocal exploration and learning (*Bottjer et al., 1984*; *Scharff and Nottebohm, 1991*; *Kao et al., 2005*; *Ölveczky et al., 2005*).

During singing, RA-projecting HVC neurons fire sparse and song-locked spike bursts that are thought to control song timing (*Hahnloser et al., 2002*; *Long and Fee, 2008*; *Ali et al., 2013*). Projection neurons in RA drive brainstem motor neurons that innervate muscles involved in singing (*Vicario, 1991*; *Wild, 1997*). Given this functional architecture, song learning has been cast as the process of transforming the timing signal in HVC into effector-specific RA motor commands by establishing connections between time-keeper neurons in HVC and muscle-related neurons in RA appropriate for producing the desired vocal output (*Figure 1D*) (*Fee et al., 2004*; *Fiete et al., 2004, 2007*; *Fee and Goldberg, 2011*).

Inputs to RA from the lateral magnocellular nucleus of the anterior neopallium (LMAN), the cortical outflow of the AFP, is thought to guide the learning process within RA by adding variability to the motor program (*Kao et al., 2005, 2008*; *Ölveczky et al., 2005, 2011*; *Thompson and Johnson, 2007*) and

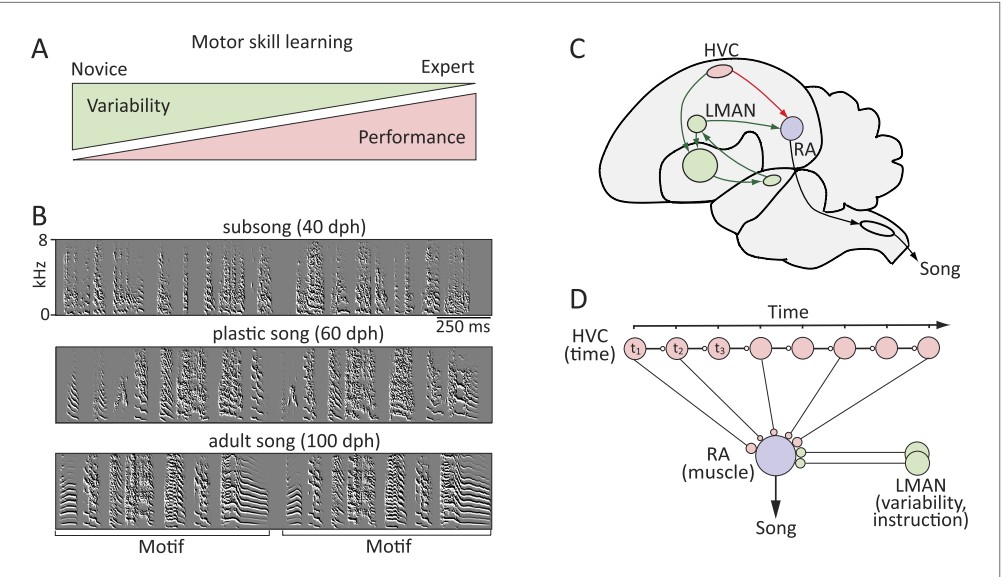

**Figure 1**. Probing the neural mechanisms underlying the regulation of motor variability in songbirds. (**A**) A strong coupling between performance improvements and variability reduction is a hallmark of most forms of motor skill learning. (**B**) Spectral derivatives of songs from a single zebra finch at different stages of song learning (dph-days post hatch) show a reduction in vocal variability as a function of learning. (**C**) The neural circuits associated with the acquisition and execution of song. HVC (red) and RA (purple) constitute the cortical part of the vocal motor pathway (VMP) and control the learned song; the anterior forebrain pathway (AFP, green), is essential for inducing vocal variability and guiding the song learning process. (**D**) Presumed functional organization of the motor pathway in which HVC represents time in the song (t) in the form of a synaptic chain network and RA neurons control specific muscles. Learning in the motor pathway is thought to be driven by plasticity in RA that is guided by input from LMAN, the output of the AFP.

mediating auditory feedback-based error-correction (*Brainard and Doupe, 2000*; *Andalman and Fee, 2009*; *Warren et al., 2011*; *Charlesworth et al., 2012*). Thus, the capacity for exploration and learning is promoted by inputs to RA from LMAN, while the consolidation of learned song occurs within the HVC-RA network.

Early in song learning, when vocal exploration is needed to drive trial-and-error learning, the RA motor program is dominated by LMAN input (*Aronov et al., 2008*; *Ölveczky et al., 2011*), but as learning proceeds and exploitation of the learned vocal patterns becomes essential for successful courtship, the temporally precise inputs from HVC take over and the premotor role of LMAN decreases (*Ölveczky et al., 2005*; *Aronov et al., 2008*).

What are the circuit-level changes that underlie this shift in premotor control and hence reduction in vocal variability? Is it age-dependent attenuation of LMAN input to RA, developmental changes in HVC input, or both? Addressing this question requires a thorough circuit-level characterization of the developmental changes in both HVC and LMAN input to RA. While prior studies have examined anatomical correlates of song development in RA and shown significant age-related changes in both the spine density of RA neurons (*Herrmann and Arnold, 1991*; *Kittelberger and Mooney, 1999*) and HVC terminal bouton frequency (*Kittelberger and Mooney, 1999*), how these anatomical changes translate into functional changes in VMP connectivity is not clear. And though the relationship between HVC fiber stimulation and evoked postsynaptic potentials in RA neurons is known to be age-dependent (*Kittelberger and Mooney, 1999*), the detailed nature of the underlying changes in circuitry has not been revealed. Similarly, our understanding of how LMAN inputs to RA neurons change during song development—essential for understanding regulation of motor variability—is lacking.

Here, we examine changes in the distribution and relative number of HVC and LMAN input to RA neurons as a function of song development. We show that while HVC-RA connections undergo significant strengthening and pruning, inputs from LMAN remain largely unchanged throughout. This suggests that LMAN's effectiveness in inducing motor variability and promoting plasticity is curtailed by

the strengthening and pruning of HVC-RA connections. A simple network model shows how strengthening and pruning of action-specific connections in motor control circuits makes the dynamics in those circuits less sensitive to external sources of variability (e.g. LMAN) and neural 'noise'. This identifies a simple and general circuit-level mechanism for learning-related regulation of motor variability.

## Results

### Estimating how inputs to RA neurons change during song development

To characterize how inputs to RA change as a function of sensorimotor learning, we used birds of ages corresponding to three distinct phases of song learning ('Materials and methods', *Figure 1B*): (1) *Subsong* (40–45 days post-hatch, dph)—the earliest stage of sensorimotor learning characterized by highly variable songs driven largely by LMAN (*Aronov et al., 2008*); (2) *Plastic song* (60–65 dph)— an intermediate stage of development characterized by recognizable but variable song elements or syllables that are subject to further change; (3) *Crystallized song* (90–130 dph)—adult stage at which a stereotyped and stable version of the bird's courtship song has developed (*Immelmann, 1969*; *Price, 1979*).

RA has two major cell types: excitatory projection neurons and inhibitory interneurons (*Spiro et al., 1999*). Adult vocalizations are driven by the precise burst firing of RA projection neurons (*Simpson and Vicario, 1990*; *Yu and Margoliash, 1996*; *Leonardo and Fee, 2005*) that are thought to be largely triggered by inputs from time-keeper neurons in HVC (*Hahnloser et al., 2002*; *Hahnloser, 2006*). Activity in RA projection neurons is further influenced by inputs from LMAN (*Ölveczky et al., 2005*; *Kao et al., 2008*), which induce variability in the RA motor program during singing (*Ölveczky et al., 2011*). This organization makes the connections between HVC and RA projection neurons a likely site of learning and memory (*Doya and Sejnowski, 1995*; *Fee et al., 2004*; *Fiete et al., 2007*) and the inputs from LMAN to RA an important locus for driving motor variability and plasticity (*Figure 1D*) (*Kao et al., 2005*; *Ölveczky et al., 2011*). Thus, our current study focuses on RA projection neurons and how their inputs from HVC and LMAN, as well as their intrinsic properties, change during learning.

Experiments were done in acute brain slices that included a large fraction of RA (*Figure 2A*; 'Materials and methods'). Because LMAN and HVC fibers enter RA in different planes (*Mooney and Konishi, 1991*; *Mooney and Rao, 1994*), we used two different slices to probe inputs from the respective nuclei, optimizing for the number of input fibers for each experiment ('Materials and methods'). Input strength was probed by stimulating the fiber tract entering RA from HVC or LMAN at different stimulation intensities (*Figure 2B*, 'Materials and methods') and recording evoked currents in RA projection neurons under whole-cell voltage clamp (*Figure 2C,D*).

Drawing on the characterization of synaptic refinement at the retinogeniculate synapse (*Hooks and Chen, 2006*; *Noutel et al., 2011*), we used the metrics of *Single Fiber* (SF) and *Maximal* (MAX) current to estimate the relative strength and number of LMAN and HVC inputs. SF currents from individual fibers were recorded at a stimulus intensity that produced a mixture of evoked unitary EPSCs of consistent peak amplitude and failures (*Figure 2E*, 'Materials and methods'). The peak of the single fiber-evoked EPSC (SF current) was used to gauge the strength of inputs from single HVC or LMAN neurons. MAX currents, estimated from the peak EPSC at saturating stimulus intensities (*Figure 2C,D*; 'Materials and methods'), were used to approximate the overall drive from HVC or LMAN fibers to single RA projection neurons.

### HVC-RA connectivity is reorganized throughout song learning

We first characterized the strength of single HVC inputs to RA projection neurons across the three age categories by recording HVC-fiber-evoked SF currents in 176 RA cells in 86 birds (*Figure 3A*). By stimulating the HVC fiber tract using an array of stimulation electrodes ('Materials and methods'), we were often able to characterize more than one SF input to a single cell (on average $1.65 \pm 0.81$; mean ± SD).

While HVC input to RA is mediated by both AMPA and NMDA receptors (*Mooney and Konishi, 1991*; *Mooney, 1992*), the NMDA component is not thought to be essential for driving song in either juvenile (*Ölveczky et al., 2005*) or adult (*Charlesworth et al., 2012*) birds, and we thus characterized the strength of HVC-RA connections by recording AMPA receptor (AMPAR)-mediated currents at a holding potential ($V_h$) of −70 mV (*Stark and Perkel, 1999*).

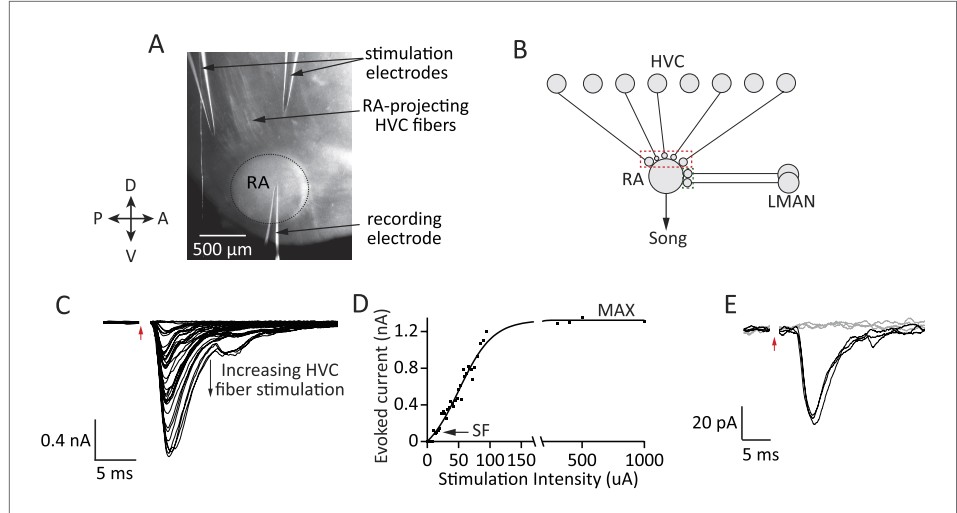

**Figure 2**. Experimental approach for probing inputs to RA projection neurons from LMAN and HVC during different stages of song development. (**A**) Bright-field image of an acute brain slice encompassing RA and incoming fibers from HVC (parasaggital slice). The fibers are stimulated and evoked currents from RA projection neurons recorded in voltage clamp. (**B**) Using different slices we can interrogate the number and strength of HVC (red) and LMAN (green) inputs to RA projection neurons. (**C**–**D**) Increasing the stimulation intensity activates increasingly more input fibers, allowing us to measure single fiber (SF) currents and maximal (MAX) currents. Examples of HVC fiber-evoked currents in a plastic-song bird. (**E**) Single fiber currents were measured at stimulus intensities that produced both failures (grey traces) and evoked EPSCs of reliable amplitude (black traces, see also 'Materials and methods'). Data from stimulating a putative single HVC fiber in a plastic-song bird. In **C** and **E**, red arrow denotes time of stimulation. Stimulus artifacts removed.

The shapes of the SF current distributions were strikingly similar to synaptic weight distributions found in other systems (***Song et al., 2005***; ***Barbour et al., 2007***), and could be well approximated by a log-normal fit for all three age groups (***Figure 3B***). SF currents shifted to higher values with age, indicating a gradual strengthening of HVC inputs throughout learning (***Figure 3B–D***). For example, while only 3% [2/72] of the SF inputs were stronger than 100 pA in subsong birds, this fraction increased to 8% [5/60] in plastic-song birds and to 21% [25/120] in adults. The median SF current also increased from 22.5 pA (IQR: 22.5, n = 72 SFs in 55 cells) in subsong birds (n = 24 birds) to 40.2 pA (IQR: 28.4; n = 60 SFs in 42 cells; p = 3 × 10$^{-7}$) in plastic-song birds (n = 23 birds) and 59.8 pA (IQR: 65.0, n = 120 SFs in 79 cells) in adult birds (n = 39 birds, p = 0.02 comparing second and third age categories, ***Figure 3D***). Consistent with a prior study (***Kittelberger and Mooney, 1999***), the input resistance of RA projection neurons did not change significantly with age (***Table 1***), an indication that our measurements of SF currents were not confounded by changes in intrinsic cell properties. The strengthening of HVC SF inputs across age categories was also accompanied by a reduction in the variability (CV) of single fiber EPSCs across repeated stimulations (***Table 1***).

In addition to stronger SF inputs, song development was accompanied by marked but non-monotonic changes in the overall input drive from HVC to RA neurons (***Figure 3E,F***; ***Table 1***). While MAX currents more than doubled from 0.55 ± 0.26 nA (n = 29 cells; mean ± SD) in subsong to 1.31 ± 0.61 nA in plastic-song (n = 24 cells, p = 3 × 10$^{-6}$), the trend reversed and MAX currents subsequently decreased by ~40% over the last month of song learning to 0.80 ± 0.43 nA in crystallized adults (n = 52 cells, p = 7 × 10$^{-4}$ as compared to plastic-song, ***Figure 3F***).

This suggests very significant pruning of HVC inputs during later stages of sensorimotor learning. To address the change in the number of functional HVC inputs to RA projection neurons in our slice, we estimated the relative number of inputs by dividing the MAX current for each cell by the age-matched mean SF current ('Materials and methods'). This revealed a 1.4-fold increase in the average number of HVC inputs during early sensorimotor learning from 18.7 ± 8.9 in subsong birds to 26.4 ± 12.2 in plastic-song birds (p = 0.01), followed by a 2.4-fold decline to 10.8 ± 5.8 in crystallized-song adults (p = 2 × 10$^{-6}$ between plastic-song and adult, ***Figure 3G***).

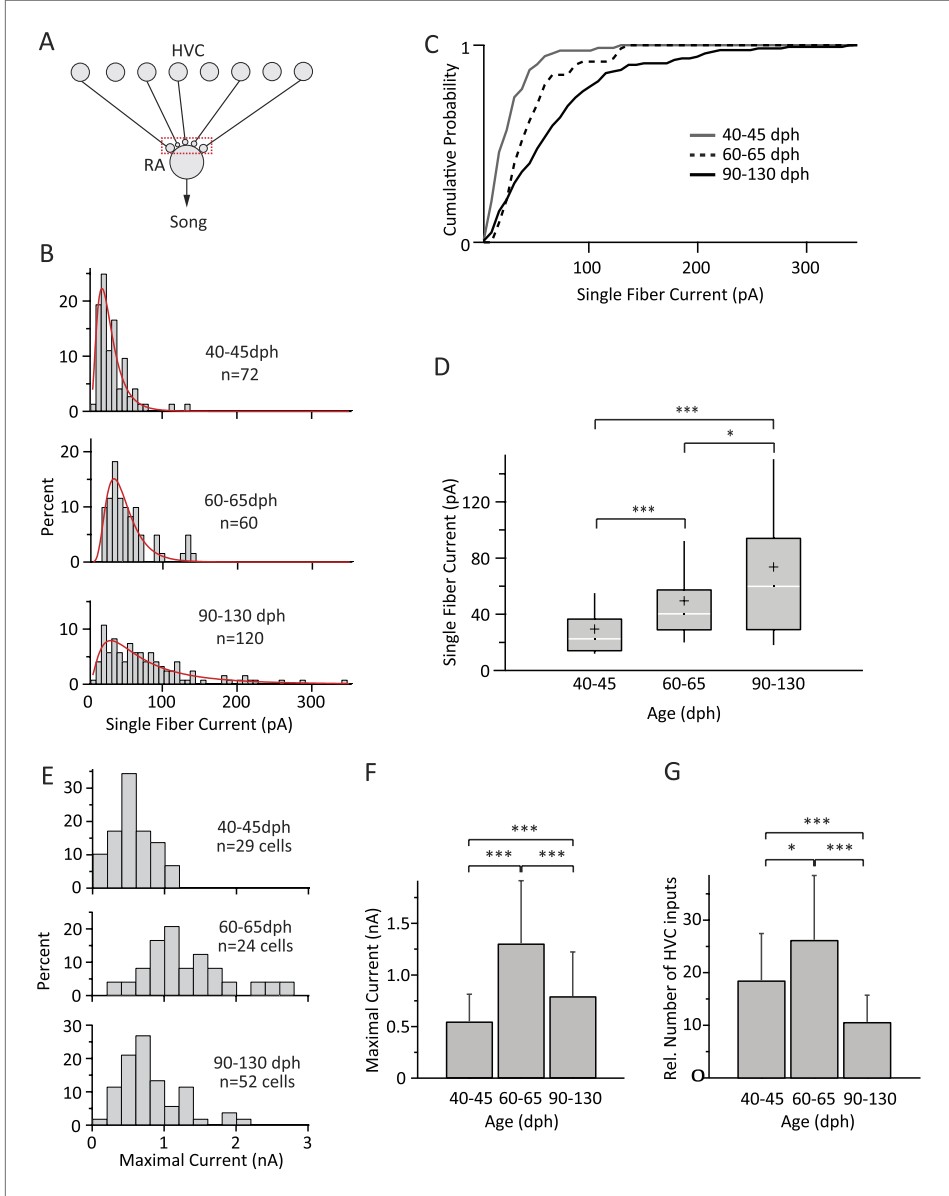

**Figure 3**. Inputs from HVC to RA neurons change throughout song development. (**A**) Schematic highlighting the inputs being probed (red box). (**B**) Distributions of SF currents for the three age groups tested. Red line represents the log-normal fit to the data ('Materials and methods'). (**C**) Cumulative SF current distributions. (**D**) Box and whisker plots showing the median (white line), IQR (grey box), mean (+), and the 10th and 90th percentile of the SF current distributions (whiskers). (**E**) Distributions of MAX currents for the three age groups. (**F**) Average MAX currents. (**G**) Average number of HVC inputs to an RA projection neuron in our slice preparation. Error bars in **F** and **G** denote standard deviations.

## LMAN-RA connectivity remains largely unchanged throughout song learning

We next examined how inputs from LMAN to RA change with learning (*Figure 4A*). Despite a reduction in the size of LMAN with development (*Bottjer et al., 1985*; *Bottjer and Sengelaub, 1989*), the number of RA-projecting LMAN neurons remains largely unchanged during sensorimotor learning (*Nordeen et al., 1992*), as does the topography of LMAN projections to RA (*Iyengar et al., 1999*). Yet whether and how the functional properties of these inputs change during song learning is not known. To address this, we employed the same experimental strategy as for HVC-RA

**Table 1.** HVC–RA synaptic properties

| Birds | Age (dph) | Input resistance (MΩ) | Capacitance (pF) | Spontaneous firing rate (Hz) | SF peak amplitude (pA) | SF CV | SF latency to peak (ms) | MAX peak amplitude (nA) | MAX CV | MAX latency to peak (ms) |
|---|---|---|---|---|---|---|---|---|---|---|
| Subsong Juvenile (24) | 40–45 | 135.84 ± 95.09 | 96.80 ± 22.77 | 6.74 ± 4.79 | 29.52 ± 2.58 | 0.30 ± 0.21 | 9.24 ± 2.94 | 0.53 ± 0.26 | 0.05 ± 0.04 | 8.57 ± 2.13 |
|  |  | (55) | (55) | (55) | (72) | (72) | (72) | (29) | (29) | (29) |
| Plastic-song Juvenile (23) | 60–65 | 123.95 ± 93.26 | 95.48 ± 27.47 | 7.25 ± 5.10 | **49.60 ± 3.92** | **0.19 ± 0.14** | 8.39 ± 2.54 | **1.31 ± 0.61** | 0.05 ± 0.04 | **7.47 ± 1.50** |
|  |  | (42) | (42) | (42) | **(60)** | **(60)** | (60) | **(24)** | (24) | **(24)** |
|  |  |  |  |  | *p < 0.001† | *p < 0.001* |  | *p < 0.001* |  | *p < 0.05 |
| Crystalized-song adult (39) | 90–130 | 105.80 ± 76.05 | **86.71 ± 24.92** | 7.95 ± 5.50 | **73.56 ± 5.49** | **0.18 ± 0.15** | 7.68 ± 2.84 | 0.80 ± 0.43 | 0.05 ± 0.04 | **7.08 ± 1.70** |
|  |  | (80) | **(80)** | (80) | **(120)** | **(120)** | **(120)** | **(52)** | (52) | **(52)** |
|  |  |  | *p < 0.05* |  | *p < 0.001† | *p < 0.001* | *p < 0.001* | *p < 0.001* |  | *p < 0.01* |
|  |  |  |  |  | **p < 0.05† |  |  | **p < 0.001* |  |  |

Values are mean ± SD *Versus Subsong Juvenile; **versus plastic-song juvenile; statistically significant differences in **bold**.

*Two-tail Student's *t* Test.

†Wilcoxon Rank–Sum Test: used when one or more of the distributions under comparison were significantly non-parametric, as determined by the Kolmogorov–Smirnov.

connections, but in coronal slices that isolated LMAN inputs to RA ('Materials and methods'), measuring LMAN-fiber evoked EPSCs at both +40 mV and −70 mV (*Figure 4B*). Input strength was characterized at the depolarized holding potential because, unlike HVC inputs, inputs from LMAN are mediated predominantly by NMDA receptors (NMDARs) (*Mooney and Konishi, 1991*; *Mooney, 1992*; *Stark and Perkel, 1999*), which are blocked by $Mg^{2+}$ at −70 mV (*Mayer et al., 1984*; *Nowak et al., 1984*).

In contrast to HVC SF inputs, the distribution of LMAN SF EPSCs did not change significantly across the three age categories (*Figure 4C,D*). The median SF peak EPSC was 138.32 pA (IQR: 202.0 pA, n = 40 SFs in 24 cells) in subsong birds (n = 7 birds), 90.73 pA (IQR: 143.5 pA, n = 45 SFs in 30 cells) in plastic-song birds (n = 13 birds), and 94.41 pA (IQR: 136.5 pA, n = 38 SFs in 33 cells) in adult birds (n = 12 birds; p > 0.1 for all pairwise Mann-Whitney-Wilcoxon test comparisons, *Figure 4E*). Though the mean strength of SF LMAN input was near identical in plastic-song and adult birds (146.1 and 142.3 pA respectively, p = 0.75), these numbers were ~25% lower than the mean SF input in subsong birds. Though this difference did not reach statistical significance (p > 0.12), we cannot rule out a slight developmental weakening of LMAN SF input early in sensorimotor learning.

There was also no significant change in the shape of the SF distributions across the different age categories (p > 0.1 for all pairwise Kolmogorov–Smirnov test comparisons). The CV of the peak EPSC amplitude for successive stimulations of the same SF (*Table 2*) was also similar across the age groups (p > 0.2 for all pairwise Student's *t* test comparisons). The ratio of peak SF EPSC at a holding potential ($V_h$) of −70 mV vs 40 mV (*Figure 4F*), however, decreased significantly in plastic-song birds (0.10 ± 0.09, n = 45) and adult birds (0.09 ± 0.08, n = 38) as compared to subsong birds (0.17 ± 0.09, n = 40, p < 0.001 for both pairwise comparisons), suggesting a small developmental decrease in the relative contribution of AMPA vs NMDA receptors at LMAN-RA synapses during early development (*Table 2*). Interestingly, assuming a reversal potential of 0 mV for AMPA and NMDA currents, our results indicate an NMDA:AMPA ratio of 9:1 in subsong birds ('Materials and methods'), consistent with a prior study (*Stark and Perkel, 1999*).

In contrast to significant developmental changes in the HVC drive to RA neurons (*Figure 3F*), the total input from LMAN to RA projection neurons did not show a significant change across the age categories we tested (*Figure 4G*). The average MAX LMAN current was 0.42 ± 0.21 nA (mean ± SD, n = 18 cells in five birds) in subsong birds, 0.48 ± 0.28 nA (n = 18 cells in seven birds) in plastic-song birds, and 0.38 ± 0.17 (n = 15 cells in five birds) in adults (p > 0.25 for all pairwise comparisons, *Figure 4F*). The ratio of MAX current to mean SF current suggests that the number of LMAN inputs to an RA cell is also largely unchanged across sensorimotor learning (p > 0.05 for all pairwise comparisons), remaining relatively sparse throughout (*Figure 4H*).

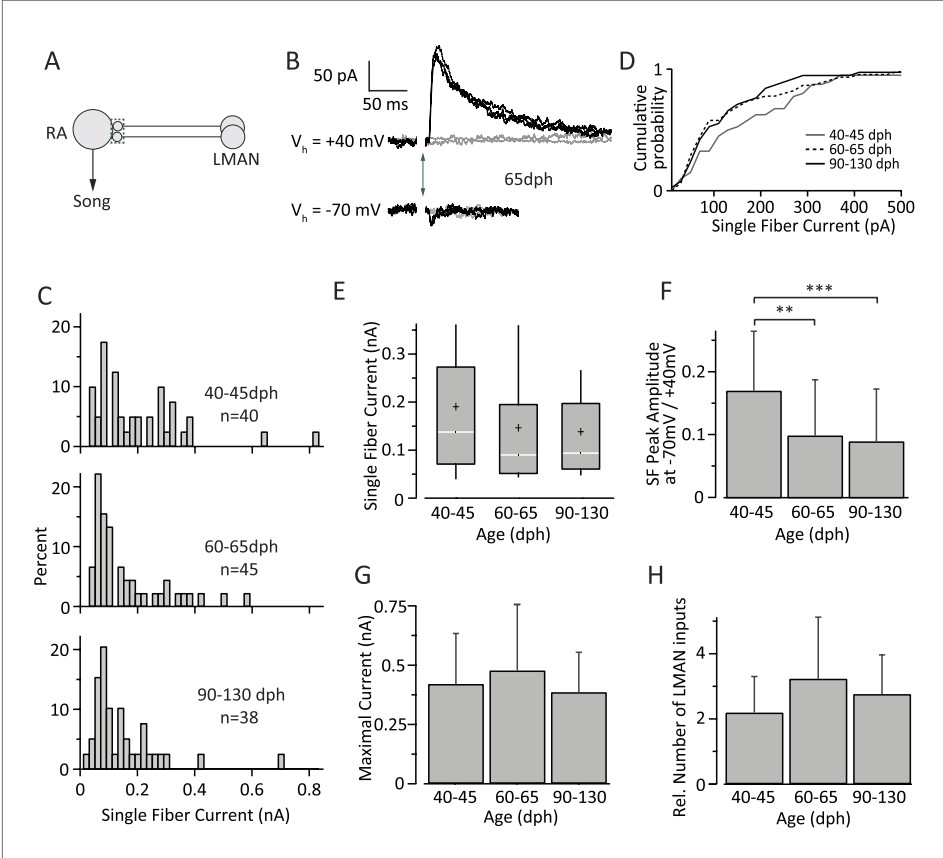

**Figure 4**. Inputs from LMAN to RA neurons remain largely unchanged throughout song development.
(**A**) Schematic highlighting the inputs being probed (green box). (**B**) Currents evoked in response to stimulating the LMAN fiber tract with an intensity resulting in either failures (grey) or EPSCs of consistent amplitudes (black). Top: at a holding potential ($V_h$) of +40 mV, where both AMPA- and NMDA receptor-mediated currents are measured. Bottom: holding potential of −70 mV, where AMPAR-mediated currents dominate. Stimulus artifacts removed. (**C**) Distributions of SF currents at $V_h$ = +40 mV for the three age groups tested. (**D**) Cumulative SF current distributions. (**E**) Box and whisker plots showing the median (white line), IQR (grey box), mean (+), and the 10th and 90th percentile of the SF current distributions (whiskers). (**F**) The mean ratio of SF currents evoked at $V_h$ = +40 mV (AMPA and NMDA receptor mediated) and −70 mV (AMPAR mediated). (**G**) Average MAX currents. (**H**) Average number of LMAN inputs to an RA projection neurons in our slice preparation. Error bars in **F**, **G**, and **H** denote standard deviations.

## Intrinsic excitability of RA projection neurons is unchanged throughout song learning

In addition to changes in synaptic strength, learning and development have been shown to produce changes in the intrinsic excitability of neurons (**Brons and Woody, 1980**; **Sourdet et al., 2003**; **Zhang and Linden, 2003**; **Zhang, 2004**). As the song-related firing frequency of RA neurons increases throughout sensorimotor learning (**Ölveczky et al., 2011**), we sought to examine whether changes in the excitability of RA projection neurons in response to depolarizing input may contribute to such changes (**Figure 5A**).

The recordings were performed in whole-cell current clamp at 35°C. This allowed us to capture the current-to-firing rate transformations in RA neurons with better control over leak currents and at more physiological temperatures, and hence higher instantaneous firing rates, than prior studies (**Mooney, 1992**; **Kittelberger and Mooney, 1999**). RA projection neurons were stimulated by injecting 0.5 s current pulses ranging in intensity from −200 pA to 2 nA in 200 pA steps (**Figure 5B**). Experiments were performed in parasagittal slices from subsong (n = 9 cells in two birds), plastic-song (n = 9 cells in two birds), and adult birds (n = 10 cells in two birds). We calculated the average instantaneous firing

**Table 2** LMAN-RA synaptic properties

| Birds | Age (dph) | SF peak amplitude Vh = +40 mV (pA) | SF CV Vh = +40 mV | SF latency to peak Vh = +40 mV (ms) | Ratio SF peak amplitude at Vh = −70 mV to that at Vh = +40 mV | MAX peak amplitude Vh = +40 mV (nA) | MAX CV Vh = +40 mV | MAX latency to peak Vh = +40 mV (ms) | Ratio MAX peak amplitude at Vh = −70 mV to that at Vh = +40 mV |
|---|---|---|---|---|---|---|---|---|---|
| Subsong Juvenile (7) | 40–45 | 191.30 ± 162.24 | 0.10 ± 0.06 | 11.06 ± 3.90 | 0.17 ± 0.09 | 0.42 ± 0.21 | 0.08 ± 0.04 | 12.97 ± 1.71 | 0.32 ± 0.28 |
| | | (40) | (40) | (40) | (40) | (18) | (18) | (18) | (18) |
| Plastic-song Juvenile (13) | 60–65 | 147.41 ± 132.10 | 0.12 ± 0.05 | 10.66 ± 4.29 | **0.10 ± 0.09** | 0.48 ± 0.28 | 0.07 ± 0.05 | 11.56 ± 0.73 | 0.28 ± 0.21 |
| | | (45) | (45) | (45) | **(45)** | (18) | (18) | (18) | (18) |
| | | | | | ***p < 0.01*** | | | | |
| Crystalized-song adult (12) | 90–130 | 141.24 ± 126.40 | 0.10 ± 0.06 | 10.18 ± 4.20 | **0.11 ± 0.08** | 0.39 ± 0.17 | **0.05 ± 0.03** | 11.23 ± 3.38 | 0.30 ± 0.17 |
| | | (38) | (38) | (38) | **(38)** | (15) | **(15)** | (15) | (15) |
| | | | | | ***p < 0.001*** | | ***p < 0.01*** | | |

Values are mean ± SD *Versus Subsong Juvenile; **Versus plastic-song juvenile; statistically significant differences in **bold**.
*Two-tail Student's *t* Test.
Wilcoxon Rank–Sum Test: used when one or more of the distributions under comparison were significantly non-parametric, as determined by the Kolmogorov–Smirnov test.

frequency (IFF) in response to three repeated current injections (**Mooney, 1992**; **Kittelberger and Mooney, 1999**) (**Figure 5C**).

RA projection neurons showed marked spike-frequency adaptation that increased with increased current stimulation (**Figure 5D,E**, R = 0.97, p = $10^{-7}$, 'Materials and methods'). The magnitude of adaptation (initial vs steady-state IFF) was similar across the different age groups (p > 0.05) for all pairwise comparisons at each stimulus intensity, a finding consistent with a previous study (**Kittelberger and Mooney, 1999**).

The input/output gain of RA projection neurons, estimated from the linear part of the F-I curve, was also similar across the age groups, consistent with previous reports (**Mooney, 1992**; **Kittelberger and Mooney, 1999**). For current injections ≤1.4 nA, the average slope of the linear fit was 163.37 Hz/nA for subsong, 185.33 Hz/nA for plastic-song, and 201.50 Hz/nA for adult birds ($R^2 ≥ 0.99$; **Figure 5F**). Though there was a trend towards a small increase in the overall gain from subsong birds to adults, this did not reach significance in our data set (p > 0.11). Additionally, IFF at a given stimulus intensity was not significantly different across age categories (**Figure 5F**, p > 0.05).

## A simple network model for learning-related regulation of variability

Previous studies have shown that song development, and hence the reduction in vocal variability, is accompanied by a relative decrease in LMAN's premotor influence (**Brainard and Doupe, 2000**; **Ölveczky et al., 2005**; **Aronov et al., 2008**; **Kao et al., 2008**) in favor of an increased role for HVC (**Aronov et al., 2008**; **Fee and Goldberg, 2011**). But the intuitive circuit-level explanation—a learning-related increase in overall HVC drive to RA coupled with a decrease in LMAN input—was not observed. On the contrary, our results revealed that the LMAN input remains largely unchanged throughout song learning (**Figure 4**), while the overall HVC drive to RA actually *decreases* (**Figure 3F**). How to account for the developmental switch in premotor drive from LMAN to HVC in light of these results? One possibility is that the strengthening and pruning of HVC-RA connections makes LMAN inputs to RA less effective.

To probe this idea further and, more generally, to examine the neural mechanisms that can regulate motor variability, we constructed a simple computational model of the HVC-RA-LMAN network (**Figure 6A**). We used this model to evaluate how song-related firing patterns of simulated RA neurons change when HVC-RA synapses are strengthened and pruned and how this process is impacted by other possible changes to the circuit, thus dissecting the potential contributions of various neural mechanisms to the regulation of motor variability.

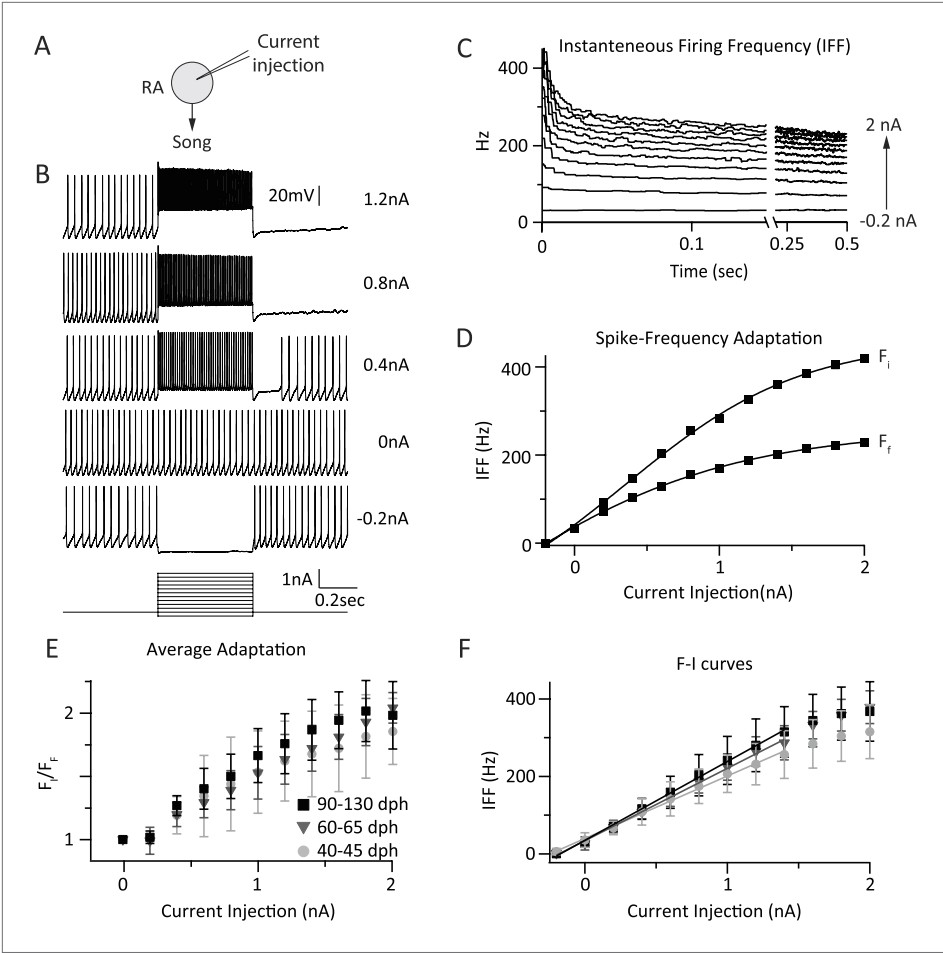

**Figure 5**. Intrinsic properties of RA projection neurons do not change significantly with song development. (**A**) To test intrinsic excitability of RA neurons as a function of age we injected current into RA cells and measured the membrane voltage in current clamp. (**B**) Membrane voltage in an RA projection neuron as a function of injected current. Example shown is from a 60 dph bird. (**C**) Average instantaneous firing frequency (IFF) throughout the 0.5-s current injection for the cell in **B** (n = 3 current sweeps). Traces correspond to differing intensities of injected current (−0.2–2 nA in increments of 0.2 nA). (**D**) Average IFF during the initial ($F_i$) and final ($F_F$) 5 ms of the current injection for the cell in **B**. (**E**) Spike frequency adaptation ($F_i/F_F$) in RA neurons for the three age categories as a function of stimulation intensity (n = 9 cells from subsong, 9 from plastic-song, and 10 from adult birds). (**F**) F-I curves for the same population of RA neurons as in **E**. p > 0.1 for all pairwise comparisons across age categories of the slope of the linear portion of the F-I curve.

The gradual strengthening and pruning of HVC inputs to RA in our model were parameterized to fit our experimental observations (*Figures 3, 4*; 'Materials and methods'). In particular, we focused on changes that occur from plastic song to adult song, as variability in both song and RA firing patterns has been quantified and seen to decrease during this phase of learning (*Ölveczky et al., 2011*). The combination of strengthening and pruning HVC input to RA led to a dramatic reduction in LMAN's capacity to drive rendition-to-rendition variability in RA neurons (*Figure 6A,B*). Despite the simplicity of the model, these changes mirrored almost perfectly what has been observed in vivo (*Ölveczky et al., 2011*), meaning that the strengthening and pruning of HVC-RA synapses can, on their own, explain much of the learning-related reduction in song variability. Our simulations also showed that strengthening and pruning of HVC-RA connections is far more effective in reducing LMAN-induced variability in RA neurons than either strengthening or pruning on its own (*Figure 6B*), suggesting that the two processes act synergistically to reduce variability.

While our experiments did not reveal any significant learning-related changes in LMAN inputs to RA, small modifications at this synapse, particularly early in learning, cannot be ruled out. To explore

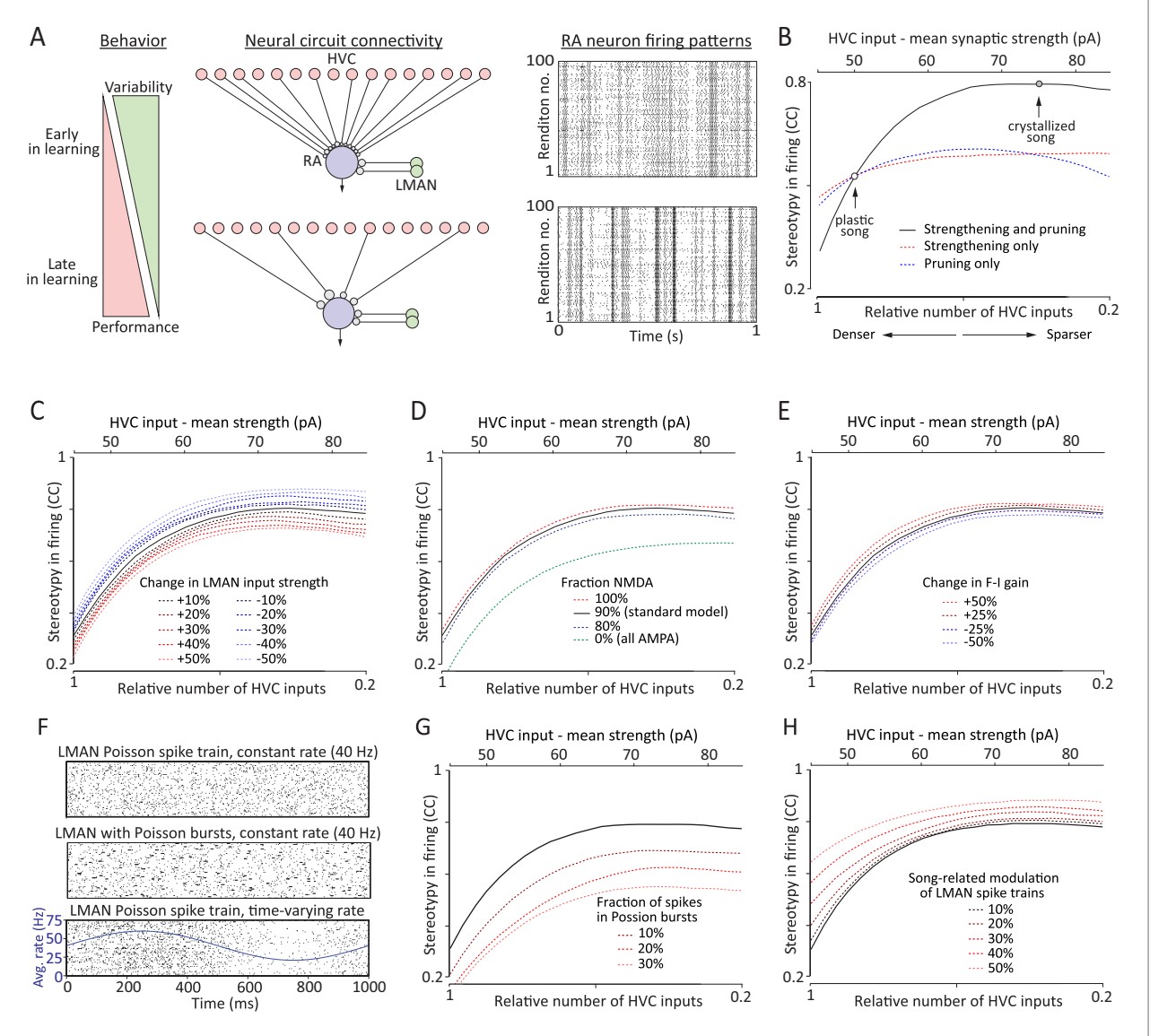

**Figure 6**. A simple model relates strengthening and pruning of connections in a motor control network (here: HVC-RA) to reduced motor variability. (**A**) Top: the early phase of song learning is characterized by high vocal variability (left panel). This is associated with relatively larger number of weaker inputs from HVC to RA neurons (middle panel). A simple model of this circuit organization ('Materials and methods'), in which RA neurons integrate input from precise HVC time-keeper neurons and variable LMAN neurons, produces variable firing patterns in RA (right panel). Bottom: song learning is associated with a strengthening and pruning of HVC-RA connections and a concomitant decrease in song variability. Pruning the relative number of active HVC-RA inputs in our model while strengthening the remaining ones (middle panel) dramatically decreases the variability in song-related RA firing (right panel). The simulations of RA spike trains were done using model parameters consistent with our experimental data from age groups 2 (top, plastic song) and 3 (bottom, crystallized song), respectively. Note that the simulations for the two neurons are independent (i.e., the 'older' is not derived from strengthening and pruning connections of the 'younger'). See 'Materials and methods' for further details. (**B**) The average cross-correlation (CC) between spike trains during different 'song' renditions of the simulated RA neuron increases with the degree of strengthening and pruning of HVC-RA synapses. The open circles correspond to the age groups 2 and 3 in our data set. Solid black line represents simulations using 'standard' model parameters chosen to conform to our experimental data (see 'Results', *Figures 3–5*, and 'Materials and methods'). Dashed lines show simulations where inputs from HVC were either only strengthened (red) or pruned (blue), relative to the 'standard' model at age group 2. The x-axis (bottom labels) shows the fraction of active HVC inputs to RA ('Materials and methods'), as well as the average strength of HVC inputs (top labels). (**C**–**E**) Effects on variability in RA firing from: (**C**) strengthening/weakening LMAN input by up to 50%, (**D**) changing NMDA:AMPA ratio at the LMAN-RA synapse, and (**E**) changing the gain (F-I relationship) of the RA neuron. All changes are relative to the 'standard' model (black lines) in 'B'. (**F**–**H**) Effects on variability in RA firing stemming from changes to LMAN firing patterns. Changes are relative to the 'standard' model (black lines). (**F**) Different LMAN spike trains tested in our model (for 50 'song renditions'). Top: Poisson spike train. Middle: Poisson spike train with 30% of the spikes in the form of Poisson bursts (see 'Materials and

*Figure 6. Continued on next page*

*Figure 6. Continued*

methods'). Bottom: time-varying Poisson firing. Blue curve shows the average instantaneous firing rate as a function of time in song with 50% modulation compared to the baseline rate. (**G**) Effect of altering the burstiness of LMAN neurons. Here, Poisson burst were added to the normal Poisson spike train. (**H**) Effect of increasing the song-locking of LMAN firing.

the consequences of such changes, we simulated both strengthening and weakening of LMAN inputs to RA relative to the 'standard' model derived from our measurements (*Figure 6C*, 'Materials and methods'). When paired with the strengthening and pruning of HVC-RA connections, however, weakening LMAN input by as much as 50% explained less than 20% of the overall decrease in the rendition-to-rendition variability of RA firing. Thus age-related changes in LMAN-RA connectivity are unlikely to play a major role in reducing song variability during learning.

Our results also suggested that the NMDA:AMPA ratio at LMAN-RA synapses increases modestly during early development (between subsong and plastic-song; *Figure 4F*). To probe whether changes in the receptor composition of LMAN-RA synapses impact LMAN's capacity to drive variability in RA, we ran simulations with different NMDA:AMPA ratios. Eliminating the AMPA component completely (100% NMDA) only decreased the variability in RA neurons by a few percent (*Figure 6D*). But even if a small fraction of the early reduction in vocal variability can be explained by a developmental decrease in AMPAR-mediated currents at the LMAN-RA synapse, this is not a plausible mechanism for reducing variability during later phases of sensorimotor learning (from plastic-song to adult) when receptor composition at the LMAN-RA synapses remains largely unchanged (*Figure 4F*).

Though we did not find any major age-related differences in the intrinsic properties of RA projection neurons (*Figure 5*), there was a slight but non-significant trend towards a steeper F-I curve with age (~10% steeper from subsong to plastic-song, and then again from plastic-song to adult birds, *Figure 5F*). Similar small changes in the gain of RA neurons, however, did not materially impact RA variability in our model (*Figure 6E*).

Our simulations thus far assumed Poisson-like input from LMAN to RA (*Figure 6F*, top panel; 'Materials and methods'), an approximation based on recordings from RA-projecting LMAN neurons in learning birds (*Ölveczky et al., 2005*). However, LMAN neurons are known to fire intermittent high-frequency bursts that may become more prominent in adults (*Kao et al., 2008*; *Kojima et al., 2013*). This raises the question of whether an age-related increase in the burstiness of RA-projecting LMAN neurons (*Figure 6F*, middle panel) could contribute to reducing vocal variability. Our simulations suggest that it does not; on the contrary, burstier LMAN spiking increases variability (*Figure 6G*; 'Materials and methods').

While song-aligned firing patterns of identified RA-projecting LMAN neurons in juvenile birds (corresponding to age groups two and three in our experiments) show no significant rendition-to-rendition correlation (*Ölveczky et al., 2005*), recordings in older adults, albeit from unidentified LMAN neurons, show significant song-locking (*Kao et al., 2008*). Not surprisingly, our simulations showed that less random (i.e., more song-locked) LMAN firing yields less variable RA firing (*Figure 6F* [bottom panel] and *Figure 6H*). Whether there is indeed a learning-related increase in the song-locking of RA-projecting LMAN neurons still needs to be experimentally tested.

Importantly, all our simulations showed that strengthening and pruning HVC input to RA decreases variability in RA firing irrespective of whether there are other concomitant changes in the circuit (*Figure 3*). Moreover, changes in HVC-RA connectivity could account for much, if not all, of the reduction in variability seen during song learning (*Ölveczky et al., 2011*). Though we cannot rule out that other mechanisms, including modifications to the spiking output of LMAN neurons (*Figure 6F–H*), LMAN-RA connectivity strength (*Figure 6C*), and the receptor composition at LMAN-RA synapses (*Figure 6D*), contribute to regulating variability, our results suggest that reorganization in HVC-RA connectivity is the dominant mechanism underlying learning-related reduction in vocal variability.

## Discussion

We set out to identify neural circuit mechanisms that regulate motor variability during motor skill learning. We used zebra finches as a model system because they learn a complex motor sequence, show very significant reduction in song variability with learning (*Ölveczky et al., 2005*; *Ravbar et al., 2012*), and have separate but converging neural pathways for controlling learned aspects of song

(VMP) and for driving vocal variability and plasticity (AFP) (*Figure 1*). Using whole-cell patch-clamp recordings in acute brain slices we interrogated how inputs from these distinct pathways to neurons in motor cortex analogue RA change during song development (*Figure 2*).

We found that HVC inputs to RA neurons increase both in strength and number early in song learning, consistent with a developmental increase in HVC innervation of RA (*Herrmann and Arnold, 1991*). However, as learning proceeds and variability is further reduced (~60–90 dph), there is a significant decrease in the number of HVC inputs and a strengthening of the remaining ones (*Figure 3*). Inputs from LMAN to RA, on the other hand, remain largely stable throughout sensorimotor learning, consistent with LMAN enabling and guiding song learning both in the juvenile and adult birds (*Leonardo and Konishi, 1999*; *Tumer and Brainard, 2007*) (*Figure 4*).

While a previous study (*Kittelberger and Mooney, 1999*) showed the relationship between HVC fiber-stimulation and evoked EPSPs in RA projection neurons becoming steeper with age, the experiments were not designed to address the circuit-level changes that underlie this trend. Ours is the first study to elucidate the detailed logic of how inputs from both HVC and LMAN to RA neurons are modified during development. The results from our study are consistent with and complement previous structural and physiological studies (*Herrmann and Arnold, 1991*; *Kittelberger and Mooney, 1999*), and together provide a comprehensive view of the structural and functional changes in RA that accompany song learning.

The degree to which these changes are driven by motor practice and learning vs genetically defined developmental programs cannot be easily parsed (*Ölveczky and Gardner, 2011*). Our results, however, establish a baseline against which the effect of targeted manipulations—behavioral, pharmacological, or otherwise—can be compared to further delineate the specific factors that contribute to synaptic and circuit-level changes in the developing song system.

## Logic of wiring in the song circuit

Our results also speak to the logic of connectivity within the song system. Because we recorded from RA neurons in slices where not all inputs from HVC and LMAN may have been faithfully preserved, the number of inputs we report (*Figures 3G and 4H*) should be seen as lower bounds. In these estimates each RA neuron receives, on average, an age-dependent 11–26 inputs from HVC and an age-independent 2–3 inputs from LMAN. The relative ratio of LMAN to HVC inputs is in agreement with structural studies, which found RA neurons to receive more synaptic contacts from HVC than from LMAN (*Canady et al., 1988*; *Herrmann and Arnold, 1991*).

That there are more HVC than LMAN inputs to RA neurons is also consistent with the hypothesized function of these inputs. If the role of LMAN inputs is to 'experiment' on HVC-RA connections, as is widely assumed (*Fiete et al., 2004*), having each RA neuron receive only a few relatively strong inputs from LMAN ensures that single LMAN neurons can drive changes in RA firing and that the interference from different 'experimenters' is limited. Inputs from HVC, on the other hand, are required to drive on average 12 bursts per song motif in adult birds (*Leonardo and Fee, 2005*), and hence more HVC inputs may be needed. Whether RA bursts are driven by single HVC neurons, or whether multiple concurrent HVC inputs are required to trigger bursting in RA remains to be explored.

## Circuit mechanisms underlying learning-related decrease in song variability

The circuit-level changes that accompany song learning and the associated reduction in vocal variability were in many ways surprising and counterintuitive. Inputs to RA from LMAN, the main source of vocal variability in juvenile birds (*Ölveczky et al., 2005*, *2011*), remain largely unchanged, while the overall input drive from HVC, which provides RA with the precise timing input that ultimately drives adult stereotyped song (*Hahnloser et al., 2002*; *Long and Fee, 2008*; *Long et al., 2010*), actually *decrease* with song development. We show that this decrease is driven entirely by the elimination of HVC inputs; remaining inputs from HVC continue to get stronger (*Figure 3*).

*Pruning* connections from HVC to RA may serve to decrease the number of depolarizing events in RA neurons, thereby reducing RA neurons' sensitivity to LMAN input. This is because NMDARs, which dominate at the LMAN-RA synapse (*Mooney and Konishi, 1991*; *Mooney, 1992*; *Stark and Perkel, 1999*), require additional depolarizing input to be effective (*Mayer et al., 1984*; *Jahr and Stevens, 1990*). Thus if AMPAR-mediated HVC inputs to RA are being eliminated with learning (*Figure 3G*), such depolarizing events will become less frequent and thereby limit the times at which LMAN's can

drive motor variability. Increased hyperpolarization of RA projection neurons over the course of development (*Ölveczky et al., 2011*) could further curtail LMAN's capacity to drive spiking in RA neurons at times when there is no other depolarizing input.

*Strengthening* individual HVC inputs to RA, on the other hand, could decrease LMAN's effectiveness by saturating RA neurons. During singing, RA projection neurons tend to fire bursts of spikes (*Leonardo and Fee, 2005*; *Ölveczky et al., 2011*), the instantaneous frequency of which increases with development (*Ölveczky et al., 2011*). This trend is likely driven, in part at least, by the increased strength of individual HVC inputs (*Figure 3*). Thus at times when RA neurons are released from hyperpolarization and NMDAR block, they will be increasingly saturated by their HVC input, thus reducing the capacity of LMAN to further modulate their firing.

A simple model inspired by the HVC-RA-LMAN network (*Figure 6*) formalized the above intuition and showed that strengthening and pruning of action-specific connections in a motor control network, such as the VMP, reduces its sensitivity to external sources of variability (e.g., LMAN). Though strengthening and pruning can contribute independently, we found that the reduction in variability is accentuated when the processes co-occur (*Figure 6B*).

Intriguingly, the reorganization of HVC-RA synapses we observed experimentally (*Figure 3*) was sufficient to explain the developmental decrease in motor variability seen in vivo (*Ölveczky et al., 2011*). We did not find any evidence that developmental changes in the strength or number of LMAN-RA connections contribute to this process (*Figure 4*). Yet even if there are modest changes, the effect on motor variability is likely to be small compared to those induced by changes at the HVC-RA synapse (*Figure 6C*). Furthermore, our simulations suggest that changes in HVC and LMAN input to RA are largely additive in terms of their effects on motor variability, suggesting independent mechanisms through which the song system can regulate variability.

We also explored whether and how changes to the firing patterns of LMAN neurons may impact variability and found that increased burstiness increases rendition-by-rendition variability in RA (*Figure 6G*). Interestingly, female-directed 'performance' song, which is associated with less bursty LMAN firing (*Kao et al., 2008*), is significantly more stereotyped than undirected 'practice' song (*Stepanek and Doupe, 2010*). This suggests that modulating the burstiness of LMAN neurons, a basal ganglia-dependent process (*Kojima et al., 2013*), could be an effective mechanism for fast and context-dependent regulation of vocal variability.

Another aspect of LMAN firing patterns that can impact song variability is the degree to which LMAN neurons are locked to song (*Figure 6H*). While activity of RA-projecting LMAN neurons in juvenile birds shows no significant rendition-to-rendition correlation (*Ölveczky et al., 2005*), recordings from non-identified LMAN neurons in adult birds have shown a relatively high degree of song-locking (*Kao et al., 2008*). How rendition-to-rendition variability in LMAN firing is regulated, and whether it contributes to learning-related changes in motor variability, remains to be explored.

Though our simulations show that several circuit-level mechanisms may contribute to regulating motor variability (*Figure 6*), our experimental and modeling results taken together suggest that strengthening and pruning of HVC-RA connections is the dominant mechanism during song learning.

## A general mechanism for learning related reduction of motor variability?

Learning-related changes in action-specific circuits, like the ones we describe (*Figure 3*), have been demonstrated also in the mammalian cortex (*Silva et al., 2009*; *Xu et al., 2009*; *Wang et al., 2011*; *Caroni et al., 2012*; *Fu et al., 2012*), raising the possibility that the mechanism for coupling motor variability and skill learning suggested by our experiments may apply more broadly, including to mammalian motor learning.

Whether there is an LMAN-equivalent in mammals, or how exploratory variability is induced and regulated in mammalian motor control circuits more generally, remains to be explored. But regardless of whether the source of motor variability is intrinsic neural noise or a dedicated LMAN-equivalent input, our simulations suggest that learning-related strengthening and pruning of action-specific connections in motor control circuitry is, under the assumptions of our model, sufficient to ensure reduction in motor variability (*Figure 6*).

Having the source of variability in motor control networks be mediated predominantly via NMDARs, as in songbirds, makes the variability in these circuits (e.g., RA) more sensitive to the learning-related reorganization of their connectivity (*Figure 6D*). Inducing variability through NMDARs also ensures

that the resulting motor exploration is focused on instances when there are other depolarizing inputs to the neuron, that is, when there are active connections in the control network that can be experimented with and modified.

## Advantages of learning-dependent regulation of motor variability and plasticity

Having learning-related reorganization of action-specific connections regulate motor variability lends considerable flexibility to the process of motor skill learning. Consider the case of learning multiple actions (e.g., syllables in a song). Assuming that learning an action leads to synaptic strengthening and pruning in the neuronal assemblies that encode and control that action (*Xu et al., 2009*; *Wang et al., 2011*; *Fu et al., 2012*), then, in the absence of other mechanisms limiting plasticity, the network should retain its native capacity to learn new actions as long as those are not contingent on network connections associated with already acquired ones. In the songbird system, a tight correspondence between specific actions (i.e., syllables) and HVC-RA connections is assured, since one HVC neuron only contributes to one time-point in the song (*Hahnloser et al., 2002*).

A specific prediction from this conceptual model as it relates to birdsong is that syllables that have not yet been fully formed (i.e., ones still far from matching the 'template') should be more influenced by LMAN, and hence be more variable, than syllables that are already a close match. Intriguingly, this is exactly what was reported in a recent study (*Ravbar et al., 2012*). The authors found a strong correlation between the distance of a syllable from its target and its variability. Thus the variability of two actions (syllables) was a function of how well they had been learned. The mechanism we propose for regulating the effect of LMAN provides a simple circuit-level explanation for this observation.

## Circuit-level mechanisms for regulating learning and plasticity during song development

The circuit-level changes we observed during song development may impact the capacity for learning beyond reducing exploratory variability. Since HVC-RA connections are a likely substrate for song learning, having an abundance of these connections, as is the case during the height of sensorimotor learning (*Figure 3G*), means that the learning system has a relatively large number of connections to explore and modify. But as the system 'learns' which HVC inputs to strengthen and which to eliminate, the number of connections decreases, thus also shrinking the substrate for plasticity and learning, making the acquired behavior more robust to change. This general motif of initial hyper-innervation, followed by synaptic strengthening and pruning is also seen in other developing circuits, including the mammalian cortex (*Huttenlocher, 1979*; *Rakic et al., 1994*; *Katz and Shatz, 1996*; *Hashimoto and Kano, 2003*; *Walsh and Lichtman, 2003*).

In summary, our study characterized circuit-level changes that accompany song development and identified a simple, general, and adaptive mechanism for coupling motor skill learning and variability in a way that does not compromise the capacity for future learning and plasticity in motor control circuits. This offers a circuit-level explanation for one of the most ubiquitous features of motor skill learning, namely the action-specific and learning-related reduction in motor variability.

# Materials and methods

## Animals

The 124 male zebra finches (*Taeniopygia guttata*) used for this study were obtained from our breeding colony. The care and experimental manipulation of the animals were carried out in accordance with guidelines of the National Institutes of Health and were reviewed and approved by the Harvard Institutional Animal Care and Use Committee. Since our aim was to characterize how inputs to RA change as a function of sensorimotor learning, which in zebra finches takes place between ~35–90 dph (*Immelmann, 1969*), we focused our experiments on birds in this age range. Our experimental subjects were divided into three age groups corresponding to three distinct stages of song learning: (i) subsong juveniles (40–45 dph), (ii) plastic-song juveniles (60–65 dph), and (iii) crystallized-song adults (90–130 dph).

## Slice preparation

Birds, kept on a 14-/10-hr light/dark cycle, were brought up from the aviary on the day of the experiment, anesthetized with isoflurane and subsequently decapitated. Brains were harvested and placed

in 4°C oxygenated (bubbled with 0.4 liters per minute 95% $O_2$/5% $CO_2$) artificial cerebrospinal fluid (ACSF) containing (in mM) 119 NaCl, 2.5 KCl, 1.3 $MgCl_2$, 0.5 $CaCl_2$, 1 $NaH_2PO_4$, 26.2 $NaHCO_3$, 11 D-glucose, in which equimolar choline chloride replaced NaCl to limit excitotoxicity. Osmolarity of all ACSF solutions was elevated to 350 mOsm with sucrose (**Bottjer, 2005**). All reagents were purchased from Sigma–Aldrich (St. Louis, MO).

Experiments were performed in 300 µm thick acute brain slices (**Mooney and Konishi, 1991**) cut using a vibrating microtome (Leica VT1000 S, Germany). For experiments characterizing HVC inputs to RA, parasagittal slices were sectioned at a 26° angle relative to the interhemispheric fissure by cutting the brain down the midline and gluing each hemisphere to platforms angled at 26° with the dorsal surface facing down. We found that this slice preserves more HVC inputs to RA than the straight parasagittal slice, yielding larger HVC fiber stimulation-evoked MAX currents. The slice containing the largest fraction of RA (diameter of nucleus in the cut plane >600 µm) and HVC afferent fibers, as judged visually through an IR-DIC microscope (**Figure 2A**), was used. For experiments characterizing LMAN inputs to RA, the optic tecta and brain stem were removed, and the hemispheres laid flat on their ventral surface. The brain was then cut in half along the coronal plane and the posterior half glued onto its freshly cut anterior face and sectioned. 300 µm coronal slices were cut and the one containing the largest fraction of RA (diameter of nucleus in the cut plane >400 µm) and afferent fibers from LMAN was used.

After cutting, slices were transferred to a chamber containing ACSF at 37°C, in which 50% of the NaCl was replaced with choline chloride. The slices were allowed to recover for 30 min, before being transferred to ACSF without choline chloride for an additional 30+ minutes before experiments commenced. Solutions in both chambers were allowed to cool to room temperature (20–23°C).

## Electrophysiology - recordings

Whole-cell electrophysiological recordings in RA were performed under IR-DIC visual guidance. Voltage-clamp recordings were carried out at room temperature (20–23°C) using electrodes of 2.5–3.5 MΩ resistance, filled with an internal solution containing (in mM): 35 CsF, 100 CsCl, 10 EGTA, and 10 HEPES, with pH adjusted to 7.32 with CsOH. 50 µM picrotoxin was added to the external solution to block fast feed-forward GABAergic inhibition (**Kittelberger and Mooney, 1999**; **Stark and Perkel, 1999**). Extracellular divalents (both $Mg^{2+}$ and $Ca^{2+}$) were elevated to 4 mM in order to dampen excitability and reduce spontaneous activity. HVC-evoked EPSCs were recorded at −70 mV in order to minimize contamination from predominantly NMDAR-mediated LMAN inputs (**Mooney and Konishi, 1991**; **Mooney, 1992**; **Stark and Perkel, 1999**) by means of hyperpolarization induced $Mg^{2+}$ block. LMAN-evoked EPSCs were recorded at +40 mV in addition to −70 mV in order to characterize both NMDA and AMPA receptor mediated components (**Figure 4B**). Current-clamp recordings were carried out at 35°C (TC-324B, Warner Instruments, Hamden, CT) with 2.5–3.5 MΩ electrodes containing (in mM): 130 K-Gluconate, 0.2 EGTA, 4 KCl, 2 NaCl, 10 HEPES, 2 Mg-ATP and 0.5 Na-GTP, with pH adjusted to 7.25 with KOH.

RA projection neurons were identified based on their spontaneous tonic activity and characteristic spike waveform recorded in cell-attached configuration prior to breaking into the cell (**Mooney, 1992**; **Spiro et al., 1999**). The data presented are from identified projection neurons.

Electrophysiological data were recorded with a MultiClamp 700B (Molecular Devices, Sunnyvale, CA) using mafPC software (courtesy of MA Xu-Friedman) and custom macros in Igor Pro (WaveMetrics, Portland, OR). Signals were digitized at 100 kHz and Bessel filtered at 10 kHz. Input resistance was measured throughout the experiment in response to a 10 ms long 10 mV hyperpolarization step. Series resistance was not compensated, but was monitored throughout the recording, and not allowed to vary by more than 10%. If it did, the cell was excluded from the analysis.

## Fiber-tract stimulation

Inputs were stimulated either with a pair of saline-filled glass electrodes (**Figure 2A**) or a pair of tungsten electrodes (MicroProbes, Gaithersburg, MD) selected from an array of four evenly spaced electrodes. Using the array allowed us to switch the electrodes across which stimulation was delivered and thus activate different parts of the fiber tract independently without moving the electrodes and risk losing the cell. Stimulation currents were delivered using an ISO-Flex Stimulator (A.M.P.I., Israel). The current pulse was 0.2 ms in width and varied in amplitude (see below).

HVC and LMAN afferent fibers enter RA along different tracts, allowing isolated activation of these inputs to RA (**Mooney and Konishi, 1991**). For HVC experiments the stimulating electrodes were

placed ~1 mm dorsal to RA on the fiber tract connecting HVC to RA (*tractus archistriatalis*) (**Figure 2A**). For LMAN experiments, the stimulating electrodes were placed ~1 mm lateral to RA along the fiber tract connecting LMAN and RA (**Stark and Perkel, 1999**).

The stimulus intensity was gradually increased from 10 μA to 1 mA. At the lowest intensities no current was typically evoked, but as stimulation intensity increased, the evoked postsynaptic currents (EPSCs) also increased in amplitude, until reaching a maximum (**Figure 2C,D**). Currents were included only if they showed a smooth rise to peak indicative of monosynaptic input. For current-clamp recordings, cells were stimulated with a 0.5 s direct current injection at intensities ranging from −200 to 2000 pA (**Figure 5B**).

## Quantifying synaptic refinement

The recorded current traces were smoothed using a 1 ms sliding window. The threshold for detecting a unitary EPSC single fiber (SF) input was two times the RMS of the signal acquired without any stimulation. The RMS (or 'noise') of our recordings was 2–5 pA, meaning that SF currents <4 pA were not registered. SF currents were measured at a minimal stimulus intensity that produced 25–75% failures (a pre-set criteria) relative to EPSCs of consistent amplitude. The SF was defined as the average peak EPSC (N > 3) evoked at that minimal stimulus intensity (**Figure 3B**). In 107 out of 264 recorded projection neurons, we were able to measure up to three SFs by stimulating across different neighboring stimulus–electrode pairs of the 4-electrode array. If two SF currents measurements were made in the same cell by stimulating contiguous pairs of stimulation electrodes, the second observation was only included if the evoked currents in the two cases differed significantly in peak amplitude (p < 0.05, Student's *t* test).

MAX currents were characterized at the stimulus intensity (MAX-stim) where the EPSC peak amplitude no longer increased despite a greater than threefold increase in stimulus intensity (**Figures 2E and 4B**). MAX current was the average peak response evoked at MAX-stim (N > 3; CV ≤ 0.2). MAX currents were recorded by stimulating across the electrode pair that evoked the largest EPSC.

The relative number of inputs to a cell (**Figures 3G and 4G**) was estimated from the ratio of the MAX current of a cell to the mean SF current for the given age-category. This calculation assumes that inputs to RA neurons sum linearly, and that the average SF input to an RA neuron can be estimated from the average SF current across the population of cells at the given age. An alternative method for approximating the number of inputs to a cell originally introduced to quantify number of inputs at the retinogeniculate synapse (**Hooks and Chen, 2006**), uses the SF current(s) recorded in a cell as the estimate of mean input strength. The fiber fraction (SF current/MAX current) for each input can then be estimated and averaged across the SFs in an age group to get the mean fiber fraction. The inverse of this quantity provides an alternative estimate for the number of inputs to cells in a given age group. We note that the two estimates yield very similar numbers. Estimating the number of HVC inputs using the mean SF current across the population as a proxy for any given cell's mean input, that is, the method we use for reporting the number of inputs in the text, yields 19, 26, and 11 inputs for age groups 1, 2, and 3 respectively, whereas the original method using fiber fractions (**Hooks and Chen, 2006**) yields 19, 28, and 8 inputs. Importantly, these numbers represent lower bounds on the average number of inputs to a cell at a given age.

## Quantifying intrinsic excitability of RA projection neurons

F-I curves were generated by calculating the instantaneous firing frequency (IFF) evoked by 0.5 s long direct current injections, ranging from −200 pA to 2 nA in 200 pA steps (**Figure 5B**) (**McCormick et al., 1985**). Current injections were 30 s apart and the sweep from −200 pA to 2 nA repeated three times. The IFFs for a given current injection were then averaged for each cell (**Figure 5C**). Spike-frequency adaptation was characterized both in terms of magnitude, by comparing the average IFF during the first and last 5 ms of the stimulus for each cell (**Figure 5D**), and kinetics, by fitting a single exponential to the first 100 ms of the decay of the average IFF curve at each stimulus intensity for each cell.

## Statistical analysis

SF data did not follow a normal distribution, as determined by the Kolmogorov–Smirnov test. For these distributions statistical significance was assessed using the non-parametric Mann–Whitney–Wilcoxon test. Box and whisker plots are shown as medians (white lines), with 25th to 75th percentile range boxes and 10th and 90th percentile whiskers. The other data were normally distributed (K–S test)

and differences across age groups were tested using the parametric Student's $t$ test. Statistical significance is indicated on graphs: *p < 0.05; **p < 0.01; ***p < 0.001. If nothing is indicated on the graphs, p > 0.05. Values reported are mean ± SD, unless otherwise noted.

## Log-normal fit to the distribution of HVC-RA inputs

The log-normal distribution has the form:

$$p(x) = \frac{e^{(\log x - \mu)^2 / 2\sigma^2}}{x\sigma\sqrt{2\pi}},$$

with parameters $\mu$ and $\sigma$ related to the mean and variance as $\langle x \rangle = e^{\mu + \sigma^2/2}$ and $\langle x^2 \rangle - \langle x \rangle^2 = (e^{\sigma^2} - 1)e^{2\mu + \sigma^2}$, respectively. The maximum-likelihood estimates of the parameters $\mu$ and $\sigma$, given $n$ observed data points $x_k$ (single fiber currents in our data set) are:

$$\hat{\mu} = \frac{\sum_k \log x_k}{n}, \hat{\sigma}^2 = \frac{\sum_k (\log x_k - \hat{\mu})^2}{n}.$$

We estimated the parameters of the log-normal distribution for each of the three age groups separately (*Figure 3*), according to the above equations.

## Computational model

We constructed a simple feed-forward version of a prior network model of the HVC-RA-LMAN circuit (*Fiete et al., 2007*) with the aim of identifying general mechanisms that contribute to regulating motor variability as a function of learning and development (*Figure 5A*). At the top of our hierarchical model is the timekeeper circuit (HVC), with neurons active only once during the 'song' (*Hahnloser et al., 2002*). These neurons, which produce the same pattern of activity on each rendition, project downstream to muscle-related neurons that drive the behavior (RA). Variability is introduced at the level of RA by an external source of variable input (LMAN).

The RA neuron is modeled as a leaky integrate-and-fire neuron with the membrane potential $V$ obeying the following equation:

$$\tau_m \frac{dV}{dt} = (V_R - V) + RI_{HVC} + RI_{LMAN} - V_{INH},$$

where $\tau_m$ = 20 ms is the membrane time constant, $V_R$ = −70 mV is the resting membrane potential, $I_{HVC}$ is the excitatory input from HVC neurons, $I_{LMAN}$ is the excitatory input from LMAN, and $V_{INH}$ is the tonic inhibitory input. $R$ = 260 MΩ is the input resistance of the RA neuron. Note that we measured the input resistance to be around 130 MΩ (*Table 1*), but to allow for two HVC inputs at any given time point in the song and to compensate for possible loss of HVC and LMAN inputs in our slice recordings, we multiplied this by a factor of two. The qualitative results we obtained, however, were not sensitive to this choice.

If the membrane potential $V$ reaches a threshold $V_{th}$ = −50 mV an RA spike is triggered and $V$ is subsequently reset to $V_R$. The membrane potential $V$ remains clamped at $V_R$ for a refractory period of 1.5 ms, after which it resumes its dynamics according to the above equation. There are $N_{HVC}$ = 100 HVC neurons in the model. Each HVC neuron produces a single burst of spikes in the 1000 ms long 'song motif'. The duration of each burst is 10 ms consisting of five spikes 2 ms apart. The onset of the burst of the $i$-th HVC neuron is $(i − 1) \times 10$ ms from the beginning of the song, such that the HVC bursts tile the whole interval of the song regularly and in a non-overlapping fashion.

The input from HVC neurons to the RA neuron obeys the following equation:

$$\frac{dI_{HVC}}{dt} = -\frac{I_{HVC}}{\tau_s} + \sum_{i=1}^{N_{HVC}} W_i^{HVC} \sum_{j=1}^{t_j^i < t} \delta(t - t_j^i),$$

where $\tau_s$ = 5 ms is the synaptic time constant, $W_i^{HVC}$ is the strength (peak EPSC) of the synapse from the $i$-th HVC neuron to the RA neuron, and $t_j^i$ is the time of the $j$-th spike produced by the $i$-th HVC neuron.

Each model RA neuron receives input from two LMAN neurons (*Figure 4G*), each with a firing rate around 40 Hz (*Ölveczky et al., 2005*). We modeled the LMAN spike train as a Poisson point processes

with a rate of 80 Hz. The input from LMAN to RA is composed of AMPA and NMDA receptor mediated components. The AMPAR component is modeled as:

$$\frac{dI_{LMAN}^{AMPA}}{dt} = -\frac{I_{LMAN}^{AMPA}}{\tau_s} + r W_{LMAN} \sum_{j=1}^{t_j' < t} \delta(t - t_j'),$$

where $0 < r < 1$ is the contribution of AMPAR mediated current at LMAN-RA synapses, $W_{LMAN}$ = 120 pA is the strength (peak EPSC) of the input from LMAN to the RA neurons, compatible with our measurements (**Figure 4E**), and $t_j'$ is the time of the $j$-th spike produced by LMAN. The NMDAR mediated component is modeled as:

$$\frac{dI_{LMAN}^{NMDA}}{dt} = -\frac{I_{LMAN}^{NMDA}}{\tau_{NMDA}} + (1 - r) W_{LMAN} \sum_{j=1}^{t_j' < t} G(t_j') \delta(t - t_j').$$

Here, $\tau_{NMDA}$ = 100 ms is the synaptic time constant (**Stark and Perkel, 1999**), and $G(t)$ the voltage-dependent modulation of the NMDA R mediated current (**Jahr and Stevens, 1990**):

$$G(t) = \left(1 + \frac{[Mg]}{3.57 mM} \exp(-V(t)/16.13 mV)\right)^{-1},$$

where $[Mg]$ = 0.5 mM is the intracellular $Mg^{2+}$ concentration. The total input from LMAN to the RA neuron is $I_{LMAN} = I_{LMAN}^{AMPA} + I_{LMAN}^{NMDA}$. Unless otherwise noted (see below and **Figure 6D**) we assumed 90% NMDA and 10% AMPA at the LMAN-RA synapse (**Stark and Perkel, 1999**).

The tonic inhibition $V_{INH}$ was set to be proportional to the HVC drive, that is:

$$V_{INH} = R_{INH} m \rho,$$

where $m$ is the mean HVC-RA connection strength, $\rho$ is the ratio of active HVC-RA input (see below) and $R_{INH}$ = 800 MΩ is the 'inhibitory input resistance', that is, the proportionality constant determining the amount of inhibition as a function of HVC input. Its high value reflects the fact that $V_{INH}$ represents the inhibitory counterpart of 100 excitatory HVC inputs. With the above set of parameters, the firing rate of the RA neuron is maintained at around 50 Hz given the HVC-RA connectivity profile specified below, which is consistent with in vivo recordings (**Ölveczky et al., 2011**).

The strengths of inputs from HVC to RA are drawn from a log-normal distribution as suggested by our experimental results (**Figure 3B**). In order to probe age-related strengthening (**Figure 6B**), the mean and standard deviation of the distribution were linearly inter- and extrapolated between the values observed for our second and third age groups. We also pruned the HVC inputs by a fraction of $1 - \rho$ (i.e., their corresponding $W_i^{HVC}$ values are set to zero) as suggested by our experimental results. In order to probe age-related pruning, we linearly changed $\rho$ from 1 to 0.2 along with changes in HVC-RA synaptic distribution (**Figure 6B**). The changes in the parameters of input strength distribution and pruning were aligned such that the points $\rho$ = 0.9 and $\rho$ = 0.37 (**Figure 6B**, open circles) represent the connectivity profile measured in our second and third experimental groups, respectively, that is, a 2.4-fold decrease in the number of active inputs, and an increase in mean input strength from 50 pA to 70 pA, with the standard deviation concomitantly changing from 35 pA to 70 pA.

## Modifying LMAN input to RA

To assess how modifications to the strength of LMAN inputs to RA would affect RA firing patterns, we increased/decreased $W_{LMAN}$ in our model by up to 50% (**Figure 6C**). A prior study showed LMAN-RA synaptic input, when averaged across juvenile and adult birds, to be composed of 90% NMDA and 10% AMPA (**Stark and Perkel, 1999**). Though we did not measure the NMDA/AMPA currents directly, our results conform to this ratio. LMAN input currents measured at +40 mV holding potential ($Mg^{2+}$ block is removed) reflect both AMPA and NMDA receptor mediated components, while, at a holding potential of −70 mV, the synaptic current is mainly mediated by AMPARs. Therefore, the ratio between peak synaptic currents given the driving force at these two holding potentials (**Figure 4F**) can be used to estimate the relative contribution of NMDA and AMPA receptor mediated currents at LMAN-RA synapses using the following formula:

$$\frac{Single\,fiber\,amplitude\,at -70mV}{Single\,fiber\,peak\,amplitude\,at +40\,mV} = \frac{g_{AMPA}\,(E+70)}{(g_{AMPA}+g_{NMDA})(E-40)},$$

where $g_{AMPA}$ is the peak conductance of the AMPAR mediated component, $g_{NMDA}$ the peak conductance of the NMDAR mediated component, and $E$ the reversal potential of the excitatory synapse. Assuming $E = 0$ mV, our data (**Figure 4F**) results in $r = \frac{g_{AMPA}}{g_{AMPA}+g_{NMDA}} = 0.09$ in age group 1 and $r = 0.06$ in age groups two and three. To probe how changes in relative receptor composition may affect RA spike patterns, we varied $r$ in the range of 0–0.2 and, in addition, simulated pure AMPAR input (**Figure 6D**).

## Modifying the intrinsic properties of model RA neurons

The slope of the F-I curve of our model leaky integrate-and-fire neuron is proportional to $1/(\tau_m\, V_{th})$. Our recordings revealed a slight developmental increase in the slope of the F-I curve of RA neurons (**Figure 5F**). To address the effect that such changes may have on the firing patterns of RA neurons, we altered the membrane time constant of the integrate-and-fire model neuron in the range of $\tau_m = 16 - 25$ ms (**Figure 6E**). This corresponds to ±20% change in the slope of the F-I curve of the neuron, compared to the $\tau_m = 20$ ms case.

## Modifying LMAN firing pattern

To explore how burstiness in LMAN neurons influences variability in RA neurons, we added Poisson bursts to LMAN spike trains while keeping the rate of the LMAN firing fixed at 40 Hz (**Figure 6F**, middle panel, **Figure 6G**). The onsets of these bursts were themselves modeled as a Poisson point process, and each individual burst event was composed of a stereotypical pattern of 5 spikes, 2 ms apart (i.e., the duration of each burst was 10 ms). Therefore, in order to generate LMAN spikes trains with the rate of 40 Hz and fraction $b$ of the spikes in bursts, the rate of the tonic Poisson spikes was set to $40(1 - b)$ Hz, and the rate of burst events was set to (40 $b$/# *spikes per burst*) = $8b$ Hz.

To explore the extent to which song-locking of LMAN firing influences variability in RA, we modulated the rate of the Poisson spike trains with a single sine wave profile along the 1000 ms interval of simulation (**Figure 6F**, lower panel, **Figure 6H**).

All the simulation results in **Figure 6** were obtained by averaging 5000 realizations of the network with the given parameters. The time constant of integration in the simulations was 0.2 ms.

## Variability measure

In order to quantify the rendition-to-rendition variability, first the RA spike train for the *i-th* rendition was converted into instantaneous firing rate $R_i(t)$ as follows:

$$R_i\left(t\right) = \frac{1}{t_{k+1}^i - t_k^i},\, t_k^i < t < t_{k+1}^i,$$

where $t_k^i$ is the time of *k-th* RA spike in *i-th* rendition. The obtained instantaneous firing rates were then convolved with a 10 ms Gaussian function, to get a smoothed firing rate function $r_i(t)$. As a measure of rendition-to-rendition variability, the correlation coefficient (CC) was calculated between firing rates $R_i(t)$ for all pairs of spike trains as follows:

$$CC = \frac{1}{N(N-1)}\sum_{i=1}^{N}\sum_{j>i}^{N}CC_{i,j},$$

$$CC_{i,j} = \frac{\left\langle \bar{r}_i(t)\bar{r}_j(t)\right\rangle_t}{\sqrt{\left\langle \bar{r}_i^{2}(t)\right\rangle\left\langle \bar{r}^{2}(t)\right\rangle_t}},$$

where the averaging $\langle .\rangle_t$ is performed over the length of renditions (*1000 ms*), $N$ is the number of renditions and $\bar{r}_i(t)$ is the mean-subtracted version of $r_i(t)$. For the simulations in **Figure 6**, the number of renditions, $N$, was 200.

## Acknowledgements

We acknowledge the contributions of Yoram Burak and Alexis Dubreuil to an earlier version of the HVC-RA-LMAN model. We thank Michael Long, Chinfei Chen, Takao Hensch, Venkatesh Murthy, Maurice Smith, Maximilian Josch, Evan Feinberg, and members of the Ölveczky lab for helpful discussions and feedback on the manuscript. This work was supported by a grant from NINDS (R01 NS066408), a McKnight Scholar Award and Klingenstein Fellowship to BPÖ, and a Swartz Foundation post-doctoral fellowship to BB.

## Additional information

### Funding

| Funder | Grant reference number | Author |
| --- | --- | --- |
| National Institute of Neurological Disorders and Stroke | R01 NS066408 | Jonathan Garst-Orozco, Bence P Ölveczky |
| Swartz Fellowship | | Baktash Babadi |

The funders had no role in study design, data collection and interpretation, or the decision to submit the work for publication.

### Author contributions

JG-O, Conception and design, Acquisition of data, Analysis and interpretation of data, Drafting or revising the article; BB, Contributed network simulations, Drafting or revising the article; BPÖ, Conception and design, Analysis and interpretation of data, Drafting or revising the article

### Ethics

Animal experimentation: The 124 male zebra finches (Taeniopygia guttata) used for this study were obtained from our breeding colony. The care and experimental manipulation of the animals were carried out in accordance with guidelines of the National Institutes of Health and were reviewed and approved by the Harvard Institutional Animal Care and Use Committee (Protocol #: 10-09).

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
