## [Decision Letter]

[Editors’ note: this article was originally rejected after discussions between the reviewers, but the authors were invited to resubmit after an appeal against the decision.]

Thank you for choosing to send your work entitled “A neural circuit mechanism for regulating motor variability during skill learning” for consideration at *eLife*. Your full submission has been evaluated by Eve Marder (Senior editor) and 3 peer reviewers, one of whom, Ronald L Calabrese, is a member of our Board of Reviewing Editors. The decision was reached after discussions between the reviewers. We regret to inform you that your work will not be considered further for publication at this stage.

The authors present an electrophysiological analysis of how connectivity between HVC and RA and LMAN and RA change during the period of song acquisition and consolidation in zebra finch. They use this model to address the general issue of how motor performance and variability interact so that during acquisition variability, which is essential for learning a new skill, is reduced as performance is enhanced. The system is particularly suitable for such analyses because variability generating (LMAN) inputs and the inputs that drive learned song (HVC) separately project to the premotor area (RA). They conclude that there is no change in the variability generating inputs (LMAN) but that the song generating inputs are pruned and strengthened during song acquisition and consolidation. They then present a simple model that shows how such a system leads to increased performance and show how NMDA inputs from the variability system enhance this process.

There are several major concerns voiced by the expert reviewers that weaken enthusiasm for the paper. First and foremost is the question of novelty. The [33] study comes to much the same conclusions about the development progression of HVC inputs to RA, albeit with less elegant electrophysiological methods. RA neuronal intrinsic properties, were also characterized by [33]. How LMAN inputs to RA progress during development has not been described and represents a significant innovation in this study. The reviewers have concerns that this part of the study is not adequately documented in the data so that statistical tests are meaningful. Considerable new experimental measurements would have to be made to satisfy this criticism. Another innovation of the present study is the modeling effort, but all the reviewers are concerned that this is not well developed or described, being more of an afterthought in the Discussion. Substantial amplification of the modeling effort (to explore whether the stated results are robust to the variability in synaptic strength demonstrated in the first part of the paper, to potential age-dependent changes in the strength of LMAN inputs, time-varying LMAN rates, etc.) would be needed to satisfy this concern.

It is *eLife's* policy to not move papers forward when they would need substantial new work to meet reviewers' concerns. In this case, it is our estimate that the amount of new work required would exceed the limits of what *eLife* considers appropriate for a revision.

*Reviewer #1*:

1) Please identify the experimental system (zebra finch) in the title to meet *eLife* requirements. For example, the title might be: “A neural circuit mechanism for regulating motor variability during song learning in zebra finches.”

2) The model should be presented in Results and more thoroughly described.

3) Please make clear why the NMDA components of HVC inputs to RA was not investigated. Is there no NMDA component at these synapses?

*Reviewer #2*:

The authors convincingly demonstrate an age-dependent change in the strength of HVC inputs to RA and use a simple mathematical model of RA neurons and their inputs to explore the consequences of these changes for motor variability. The results are interesting and the writing is very clear; however I think that several significant issues must be addressed before the paper is acceptable for publication.

1) Figure 3 convincingly demonstrates significant age-dependent changes in the strength of HVC-to-RA connections. However, two other key claims made by the authors—the apparent lack of such changes in LMAN-to-RA synapses shown in Figure 4 and the claim that the intrinsic properties of RA neurons do not change, supported by data in Figure 5—are based on the absence of a statistically significant effect. Although claims about changes in neural/synaptic populations must necessarily be made based on a limited sampling of neurons, I am concerned by the fact that the non-significant findings (Figures 4 and 5) are based on smaller datasets than the reported significant finding (Figure 3). For example, total number of data points shown in Figure 3 (72+60+120=252) is more than twice as great as those in Figure 4 (40+45+38=123). Is the failure to detect a significant difference in the LMAN-RA synapses simply a result of the authors collecting less data than they obtained for HVC-RA synapses? Of even more concern is the argument based on the data shown in Figure 5. As the authors acknowledge, there is a trend towards an age-dependent effect which does not reach significance, however the total number of data points is small (n=9-10 for each condition); I am therefore not convinced that no age-dependent trend exists. This issue might be addressed in two different ways (ideally in both). First, the authors could devise an analysis to demonstrate that the difference between the LMAN-RA and HVC-RA results is not an artifact of dataset size. Second, the authors could expand their modeling approach to demonstrate that their argument about the consequences of changes in HVC-RA synapses holds true, even if the (small, and in the present analysis not statistically significant) changes in LMAN-RA connectivity and intrinsic RA properties hinted at in Figures 4 and 5 were in fact present.

2) The authors compute the relative number of synaptic inputs using a statistic based on that used by [26]. However there are several important (and potentially problematic) differences between that earlier analysis and the one used here. Hooks and Chen calculated the ratio of single fiber current divided by the maximum current (SF/MAX), whereas the authors of the paper under review compute the reciprocal of this quantity (MAX/SF). Hooks and Chen's approach prevents large biases from low single fiber currents that would make the MAX/SF ratio quite large. After averaging this ratio for all measured single fibers, Hooks and Chen then invert it to estimate relative number. Importantly, Hooks and Chen calculate this statistic for each measured single fiber current ‘individually’, whereas the paper under review takes every maximum current and divides it by the ‘average’ single fiber current for that age group. It seems like SF/MAX should first be calculated for each single fiber input, as in Hooks and Chen, and *then* averaged. Moreover, the distributions shown in Figures 3 and 4 are skewed to the right, making the mean a poor choice for calculating the relative number. Additionally, as many readers will not be familiar with this analysis technique the pros and cons of this approach should be discussed in greater detail to clarify the underlying assumptions.

3) The modeling section should be expanded and moved from the Discussion into the Results section. In the current study, LMAN input is modeled as a single Poisson input with a fixed rate, although the firing rate of LMAN neurons, while variable, is modulated over the course of a motif. Are the conclusions based on the model robust to RA neurons receiving multiple, time varying inputs from LMAN? I suspect that the answer is yes; however demonstrating this would make the model results more convincing. Also, it is surprising that the authors do not parameterize their model with the distribution of single-fiber synaptic strengths presented in the first part of the paper (the fact that model inputs are expressed in units of mV makes it hard to assess whether the model distribution is similar to the distribution observed experimentally). Would the key model results hold true using a more realistic distribution of input strengths?

*Reviewer #3*:

Performance variability can facilitate motor learning and often declines as a skill is perfected. The neural mechanisms that account for this transition from high to low performance variability as a skill is acquired are not well understood. Here the authors addressed this issue by studying developmental changes in synaptic inputs onto song motor (RA) neurons from two different sources: HVC, a song premotor nucleus that provides precise timing signals for song patterning; and LMAN, another premotor nucleus that is thought to function as a “noise” source to drive song variability necessary to song learning. The authors make whole cell recordings from RA neurons in brain slices from male zebra finches at three canonical time points during song learning, and electrically stimulate either HVC or LMAN axons to measure the strength of single fiber (SF) inputs and also to estimate the (relative) maximum (MAX) numbers of synapses from each of these sources. Using these approaches, they determine that HVC synapses become stronger and relatively less abundant over song learning, whereas LMAN inputs remain stable in strength and relative numbers; they also show that the intrinsic properties of RA neurons are also stable over this period. They then provide a simple circuit model that supports the idea that differences in how HVC and LMAN synapses onto RA neurons change with development could account for the decrease in song variability that accompanies song learning.

Overall, the experiments are well done and the manuscript is clearly written. The two major concerns relate to relative novelty and the degree to which the proposed circuit mechanism was tested.

Regarding novelty: prior anatomical and physiological studies have documented changes in the relative strength and numbers of HVC synapses on RA neurons over song learning in zebra finches, and also had characterized the intrinsic properties of RA neurons over a similar time course. Thus, the characterization of LMAN synapses on RA neurons over song learning is the novel experimental data set provided here; the modeling study is also novel. Whether these together provide enough of an advance and will appeal to a broader readership is hard for me to assess. As a songbird researcher, I find the results very interesting and they help to articulate how the consolidation of HVC synapses on RA neurons could act as a counterweight to suppress noise from LMAN inputs. I would be interested to hear what the other reviewers' thoughts are about this.

Regarding mechanism-testing: the adult songbird's capacity to re-engage vocal plasticity and variability mechanisms through deafening or other perturbations of auditory feedback could more directly test the idea that the relative strength of HVC synapses is the major regulator of LMAN's ability to influence song variability. Whether the authors are still in a place to execute any more slice experiments is unknown, but an additional manipulation of this sort would strengthen support for their conclusions.

[Editors’ note: what now follows is the decision letter after the authors submitted for further consideration.]

Thank you for sending your work entitled “A neural circuit mechanism for regulating motor variability during skill learning” for consideration at *eLife*. Your article has been favorably evaluated by Eve Marder (Senior editor) and 3 reviewers, one of whom is a member of our Board of Reviewing Editors.

The following individuals responsible for the peer review of your submission have agreed to reveal their identity: Ronald L Calabrese (BRE), Richard Mooney, (reviewer), and Samuel Sober (reviewer).

The Reviewing editor and the other reviewers discussed their comments before we reached this decision, and the Reviewing editor has assembled the following comments to help you prepare a revised submission.

The authors present a thorough electrophysiological analysis of how connectivity between HVC and RA and LMAN and RA change during the period of song acquisition and consolidation in zebra finch. They use this model to address the general issue of how motor performance and variability interact so that during acquisition variability, which is essential for learning a new skill, is reduced as performance is enhanced. The system is particularly suitable for such analyses because variability generating (LMAN) inputs and the inputs that drive learned song (HVC) separately project to the premotor area (RA). They conclude that there is no change in the variability generating inputs (LMAN) but that the song generating inputs are pruned and strengthened during song acquisition and consolidation. They then present a simple model of the system that reflects the changes in HVC and LMAN inputs to RA projection neurons observed during song maturation (learning). This model is carefully described and appropriate to the question at hand. The model is systematically explored and the modeling data supports their major conclusion that strengthening and pruning of HVC RA connections is the dominant mechanism that leads to decreased variability in performance and suggests how NMDA inputs from the viability system may enhance this process.

The paper is in general well written and clear, and the Methods are sufficiently detailed. The illustrations are clear and show all relevant data. The experiments are careful and the data convincing. The Discussion is illuminating and sets the work in a general context of motor learning that should be broadly interesting to the readers of *eLife*.

Detailed critiques by the expert reviewers include several points, which should be addressed by the authors in revision, but which do not require further experiments.

*Reviewer #2*:

In their revised submission the authors have significantly improved their manuscript by addressing some technical concerns regarding the analysis of their physiological data and by greatly expanding the scope and impact of their modeling work. In making these improvements (primarily the latter), the authors have convinced me. Although prior studies have investigated the issues they take on, the present manuscript represents a significant advance. Therefore, although it was not clear whether the original submission had sufficient impact for publication in *eLife*, I think that the revised manuscript represents a major contribution to the field and I think that it should be published. However there are a few things I would like to see addressed first:

1) The expanded modeling results greatly strengthen the authors' argument, however I think the presentation could be made a bit more clear. First, in Figure 6, the accompanying legend, and the text in Results, I suggest that the authors more explicitly specify the baseline value relative to which the changes in LMAN input, FI gain, etc. are performed. For example, the legend to Figure 4 reads: “…(c) strengthening/weakening LMAN inputs by up to 50%”. This should specify: “… up to 50% relative to the default value of W_LMAN, which was based on our empirical measurements of single-fiber currents (see Results; Figure 4)”. Doesn't need to be this exact text of course, but for all parameters varied in Figure 6 this type of information should be made more explicit. Also, if I'm reading Figure 6 correctly, the black lines in 6C–E, G and H all represent the “default” model, this should also perhaps be made explicit in the legend. Finally, the lightest red dashed lines are nearly invisible in 6G and H. Perhaps these could be darkened to enhance readability.

2) In the Discussion section the authors consider how elimination and strengthening of HVC inputs to RA could separately contribute to a reduction of sensitivity to LMAN drive. However, the modeling studies never dissociate these two variables (number vs strength of HVC inputs), although it seems that the authors are in a good position to do so. This section of the Discussion is therefore somewhat unsatisfying and highlights that the authors have not performed an important variation on their analysis that could yield additional insight. I think that the authors should either explore this directly using the model (by dissociating input sparsity from input strength to back up their qualitative assertions) or remove this consideration from the Discussion.

*Reviewer #3*:

I thoroughly enjoyed reading the greatly improved manuscript and the authors' response to the reviewers' comments, which largely addressed my concerns. This is an important study that goes to significant lengths to describe a developmental change in functional connectivity that could account for how motor variability decreases with skill learning. Overall, I believe that the study will appeal to a wide audience and serve as an extremely useful resource for those interested in neural mechanisms of motor variability and learning.

1) I was left to wonder whether the NMDA EPSC time course (i.e., decay kinetics) measured for LMAN inputs onto RA neurons are able to contribute to variability in RA neuronal firing in response to drive from HVC? The decay kinetics of NMDA-mediated currents at the LMAN to RA synapse are known to become faster with age, similar to developmental changes described at other vertebrate central synapses, especially in the mammalian neocortex. Although these changes are most often associated with sensory learning of various kinds, it would be useful to know whether a slower time course in the LMAN to RA synaptic current has an effect on RA spike variability.

2) I appreciate the authors' clarification of the novelty of their findings. I certainly agree that theirs “… is the first study to describe the functional changes in the inputs to RA from both HVC and LMAN during song development…”, and for that scope I believe the study is important and not merely incremental. And the detailed explanation they provided for why they view their characterization of the developmental changes at the HVC to RA synapse was also helpful, although I still maintain that those changes could be at least partly inferred from earlier structural and functional studies by Hermann, Kittelberger and Canady. This is not to minimize the current study's importance, but I do believe the authors could go further in integrating the earlier work into their present writing to better highlight exactly what is new here and also to more accurately place their study in the published literature.

With due respect to the authors' different perspective, I see pretty clear evidence of functional strengthening and consolidation at the HVC to RA synapse from those earlier studies when they are viewed collectively: over development, HVC bouton number in RA declines, spine density of RA projection neurons describes an inverted “U,” and the functional efficacy of the HVC to RA synapse increases. Further, Canady's serial EM reconstruction work (albeit in canary) revealed that HVC synapses are mostly (>90%) onto spines, whereas LMAN synapses are split between spines and shafts; because HVC synapses far outnumber LMAN synapses on single RA neurons (I believe the ratio was greater than 20:1, but it's been awhile since I have read it thoroughly), Canady concluded that the vast majority of inputs onto spines of RA dendrites reflected input from HVC and from other RA neurons and not from LMAN. I think it is reasonable to assume from this that the developmental changes in RA spine density reported by Kittelberger mostly reflect reorganization from those two sources and not from LMAN. This postsynaptic change is paralleled by a reduction in the frequency of putative HVC axonal boutons in RA; taken together, this seems to provide pretty clear structural evidence of a reduction in the number of HVC to RA synaptic connections with development. Lastly, those structural changes are accompanied by increased efficacy in the HVC to RA synapse, as measured by the stimulus current, EPSP onset slope relationship, which steepens significantly between 45 d juveniles and adults. At face value, this suggests to me that the HVC to RA synapse undergoes developmental consolidation in a process that involves synaptic pruning and strengthening.

Regarding determination of locus: Kittelberger's measurements of onset slope largely avoided the potential confound of polysynaptic components. He also found that spontaneous EPSC amplitudes increased (with LMAN lesions, which mimicked the developmental effects), making it less likely that the change in the stimulus-response relationship simply reflected changes in HVC fiber excitability. He noted a reduction in HVC bouton frequency along with decreased spine density and increased stimulus-response relationship. In aggregate, these findings strongly suggest to me a developmental process in which HVC synapses are pruned and strengthened, thus partially anticipating one aspect of the current study. Nonetheless, the current study goes well beyond this by providing more direct evidence of such consolidation while also providing an exact estimate of the numbers of inputs at a greater range of developmental points.

I leave it up to the authors to decide whether they see room for improvement in their discussion on these points. In my view, it takes nothing away from the current study to do so, and I do believe that a more balanced discussion of the relationship of the current study to this earlier work would be an improvement.

---

## [Author Response]

We were very pleased that the reviewers found our results interesting, and the manuscript clear and well written. While the comments we received were generally very insightful and constructive, there was an unfortunate misreading of prior work that raised questions about the novelty of our study, a concern that dampened the reviewers’ enthusiasm for our paper.

Below we show that our results could neither have been predicted nor inferred from published work. Importantly, the main conclusion of the paper, how learning‐ related strengthening and pruning of action‐specific connections in a motor control network allows motor variability to be reduced, could not have been arrived at without the new data we provide.

We thank the reviewers for their insightful and constructive comments, which have resulted in a much-improved manuscript. A point-by-point response follows.

Reviewer #1:

*1) Please identify the experimental system (zebra finch) in the title to meet eLife requirements. For example, the title might be: “A neural circuit mechanism for regulating motor variability during song learning in zebra finches*.*”*

We thank the reviewer for this suggestion, which we have adopted verbatim.

*2) The model should be presented in Results and more thoroughly described*.

We have significantly expanded the model and have now included it in the Results section.

*3) Please make clear why the NMDA components of HVC inputs to RA was not investigated*. *Is there no NMDA component at these synapses?*

Though there are NMDA receptors at HVC‐RA synapses, HVC drives singing predominantly through AMPA receptors. Blocking NMDA receptors in RA has no discernible effect on the HVC‐driven stereotyped core song. This is true both in juvenile (50) and adult birds (15). Hence, we used AMPA currents to probe the functional strength of HVC input to RA. LMAN, on the other hand, exerts its effect on RA neurons predominantly through NMDA receptors (45). This is now clarified in the text.

Reviewer #2:

*1)*
Figure 3
*convincingly demonstrates significant age-dependent changes in the strength of HVC-to-RA connections. However, two other key claims made by the authors—the apparent lack of such changes in LMAN-to-RA synapses shown in*
Figure 4
*and the claim that the intrinsic properties of RA neurons do not change, supported by data in*
Figure 5*—are based on the absence of a statistically significant effect. Although claims about changes in neural/synaptic populations must necessarily be made based on a limited sampling of neurons, I am concerned by the fact that the non-significant findings (*Figures 4 and 5*) are based on smaller datasets than the reported significant finding (*Figure 3*). For example, total number of data points shown in*
Figure 3
*(72+60+120=252) is more than twice as great as those in*
Figure 4
*(40+45+38=123). Is the failure to detect a significant difference in the LMAN-RA synapses simply a result of the authors collecting less data than they obtained for HVC-RA synapses? Of even more concern is the argument based on the data shown in*
Figure 5*. As the authors acknowledge, there is a trend towards an age-dependent effect which does not reach significance, however the total number of data points is small (n=9-10 for each condition); I am therefore not convinced that no age-dependent trend exists. This issue might be addressed in two different ways (ideally in both). First, the authors could devise an analysis to demonstrate that the difference between the LMAN-RA and HVC-RA results is not an artifact of dataset size. Second, the authors could expand their modeling approach to demonstrate that their argument about the consequences of changes in HVC-RA synapses holds true, even if the (small, and in the present analysis not statistically significant) changes in LMAN-RA connectivity and intrinsic RA properties hinted at in*
Figures 4 and 5
*were in fact present*.

This is an important point and well taken. As the reviewer suggest in the above paragraphs, even a small effect, if real, can be made statistically significant by increasing the number of observations. This is illustrated by a power calculation we did assuming log‐normal distributions and a CV of 0.88 (taken from our LMAN data) (Figure 7). The smaller the difference in the means of two distributions (means ratio closer to 1), the larger the sample size (n) required to detect a statistically significant effect. Our current sample size (∼40) is powered to detect means ratios<0.65 between two distributions with a power greater than 0.8. The means ratio between age group 1 (40 dph) and 2 (60 dph) was 0.75 (p=0.12). If we increased our sample size 2.5 fold to ∼100 for each group we would be powered to detect this ratio as a significant effect.

We also ran simulations assuming log‐normal distributions for the LMAN‐stimulation evoked SF currents (Figure 7). We estimated these distribution for each age group from the log‐normal fits to the recorded SF distributions, and ‘sampled’ from them randomly, using different number of samples. Increasing the sample number did not result in a significant effect when comparing age groups 2 and 3 (Figure 7). This was perhaps not surprising since the effect size in our data set was very small (means ratio = 0.97; p‐value = 0.75). Thus we feel confident in saying that LMAN input to RA remains largely unchanged from 60 dph into adulthood. We note that despite no major change in LMAN drive to RA during this period, there is a very significant decrease in song variability (52). We show that this decrease can be explained by developmental changes in HVC‐RA connectivity (Figure 8).Author response image 1.A. Power calculation showing required sample size as a function of effect size to demonstrate significance at p<0.05 level with a power>0.8 assuming log‐normal distributions and a CV of 0.88 (from LMAN data). Green arrow and dashed line denote the means ratio (0.75) between age groups 1 and 2 in our LMAN dataset. Means ratio between age group 2 and 3 was 0.97. To power the experiments to detect a mean ratio of 0.75 would require ∼100 samples/group. Blue dashed line shows the means ratio we are currently powered to detect (0.65). B. Simulated experiments assuming log‐normal distributions (estimated from fits to our data). We drew randomly from the probability distributions for all three age groups and plotted the fraction of simulations that showed a statistically significant difference between age groups (p<0.05). These simulations show that increasing the sample size by a factor of two, assuming a means ratio of 0.75, may end up (with ∼60% probability) showing a statistically significant effect between age groups 1 and 2

As suggested by the power calculation above (Figure 7), the simulations revealed that increasing the number of samples for LMAN‐evoked SF currents in age groups 1 and 2 may end up showing a statistically significant (p<0.05) difference in mean strength across the age groups. While the sample size (n) for our LMAN recordings is already well above the norm for slice physiology experiments in songbirds, which tend to be in the tens (e.g. [33]; Sizemore and Perkel, 2011; [66]; Wang and Hessler, 2006), we agree with the referee that it makes sense to track down small effects if they materially influence the conclusions we draw. Would a modest decrease in LMAN input strength (if real) challenge the conclusion that strengthening and pruning of HVC‐RA connections reduces motor variability? To test this we followed the reviewer’s suggestion and expanded our model to probe the effects of changing LMAN input strength by up to 50% each way (Figure 8).Author response image 2.Strengthening and pruning of HVC‐RA synapses leads to reduced variability, and this trend is robust to changes in either LMAN input strength (A) or intrinsic properties or RA neurons (B). Both panels show average pair‐wise cross‐correlation (CC) between spike trains for different ‘song’ renditions in our model network. The x‐axis shows the mean synaptic strength of HVC input to RA (top) and the relative sparseness of HVC input to RA (bottom: 0 – all HVC neurons connect to the RA neurons, but weakly. 1 – a single, strong input from HVC). The model was parameterized to conform to the experimental data (white/grey circle denotes model parameters matching the observations in age groups 2 and 3, respectively). The strength of LMAN input was kept constant throughout. Tonic inhibition onto RA neurons was assumed proportional to the HVC drive. These parameters kept the firing rates of model RA neurons around 50 Hz, i.e. in their normal range during singing (52). Note that the total drive from HVC to RA (which drives stereotyped firing) actually decreases as HVC inputs sparsen (LMAN, the variability inducing input stays the same). Thus, the reduced variability is due to a reorganization of HVC‐RA synapses, not a change in total input drive.

As is evident from these simulations (Figure 8), having developmental changes in LMAN input would not affect our conclusion that reorganization in HVC‐RA connectivity regulates variability during learning. Even a 50% decrease in LMAN strength (a gross overestimate based on our data), when accompanied by strengthening and pruning of HVC‐RA synapses (now parameterized to fit our experimental observations), only explains ∼20% of the total change in variability. The rest (∼80%) is accounted for by changes in HVC‐RA connectivity.

This simulation also suggests that variability reduction that may come from reduced LMAN input strength is largely independent of the variability reduction that result from strengthening and pruning HVC‐RA synapses. This is an interesting observation since it suggests that variability in RA can be regulated in (at least) two independent ways: (i) by changing overall LMAN drive, and (ii) by reorganizing HVC‐RA connectivity. Importantly, it means that the conclusion of the paper, i.e. that strengthening and pruning of action‐specific connections in motor control circuit regulate variability, is robust to possible developmental changes in LMAN‐RA connection strength.

We also tested whether changes in the F–I curves of RA neurons would impact our conclusions and show that, as the reviewer predicted, they do not (Figure 8).

Beyond the expanded model, which now includes analyses of how possible developmental changes to LMAN‐RA connections and intrinsic properties of RA neurons may impact variability, we now acknowledge the possibility that there may be modest age‐related changes in LMAN input early in learning (between age group 1 and 2). We show that while such changes are likely to affect variability, those effects are largely independent (i.e. additive) of the significantly larger effects resulting from the developmental reorganization of HVC‐RA connections. Our finding that the learning‐related reduction in variability can be accounted for by the strengthening and pruning of HVC‐RA connections thus remains firm.

*2) The authors compute the relative number of synaptic inputs using a statistic based on that used by*
[26]*. However there are several important (and potentially problematic) differences between that earlier analysis and the one used here. Hooks and Chen calculated the ratio of single fiber current divided by the maximum current (SF/MAX), whereas the authors of the paper under review compute the reciprocal of this quantity (MAX/SF). Hooks and Chen's approach prevents large biases from low single fiber currents that would make the MAX/SF ratio quite large. After averaging this ratio for all measured single fibers, Hooks and Chen then invert it to estimate relative number. Importantly, Hooks and Chen calculate this statistic for each measured single fiber current ‘individually’, whereas the paper under review takes every maximum current and divides it by the ‘average’ single fiber current for that age group. It seems like SF/MAX should first be calculated for each single fiber input, as in Hooks and Chen, and* then *averaged*.

We used the quantity of MAX/mean(SF) to estimate the relative number of inputs to a cell because we believe it is a better and less noisy estimate than the alternative. The assumptions underlying our choice were not made clear in the manuscript and this has now been remedied. We assume, as does Hooks and Chen, that MAX currents reflect the linear sum of all SF inputs reaching the cell, and that the one (or in our case sometimes two) SF measurement we make in a cell is a less reliable measure of the average strength of the cell’s inputs than the average strength of all inputs to all cells in the given age category.

In the case of the retinogeniculate synapse (26), all but a few inputs (∼3) are eliminated, with the remaining inputs being strong and showing limited variability. In contrast, HVC to RA connection are both more numerous and vary more in strength, making the fiber fraction a less reliable estimate of average input strength. This is why we used MAX/mean(SF) to estimate the relative number of inputs to a given cell.

However, this is not a crucial choice. If we instead estimate the fiber fraction a la Hooks and Chen for each SF, average these together, then invert the number to estimate the average number of inputs, we get very similar numbers. In fact, the trend we report on (hyper innervation followed by pruning) is slightly more pronounced (relative number of inputs for age categories 1, 2 and 3 = 19, 28, 8 respectively, compared to 19, 26, and 11 using our preferred method).

While we prefer to use MAX/mean(SF) to estimate the relative number of inputs to a cell, we wholeheartedly agree that the underlying assumptions should have been clearer. In addition, we now also quote the numbers with the alternative calculation.

*Moreover, the distributions shown in*
Figures 3 and 4
*are skewed to the right, making the mean a poor choice for calculating the relative number*.

We don’t believe that the distribution being skewed changes how best to estimate mean input strength. We are not estimating the most likely input strength, but the mean input strength across the population of inputs.

*Additionally, as many readers will not be familiar with this analysis technique the pros and cons of this approach should be discussed in greater detail to clarify the underlying assumptions*.

We completely agree that the assumptions should have been made clearer to avoid any misunderstandings. This includes the assumption that inputs sum linearly, and that we estimate the average input to a cell from the average strength in the population. We now clarify these underlying assumptions in the text.

*3) The modeling section should be expanded and moved from the Discussion into the Results section*.

We have now significantly expanded the modeling part of the paper, largely in response to the reviewers’ excellent comments (see Figures 8 and 9). The model has been moved to Results, where we agree it belongs.

*In the current study, LMAN input is modeled as a single Poisson input with a fixed rate, although the firing rate of LMAN neurons, while variable, is modulated over the course of a motif. Are the conclusions based on the model robust to RA neurons receiving multiple, time varying inputs from LMAN? I suspect that the answer is yes; however demonstrating this would make the model results more convincing*.

The only recordings (that we know of) from identified RA‐projecting LMAN neurons during undirected singing show no significant song‐locking (50). While other reports in the literature show examples of time‐varying LMAN firing (e.g. [32]), those recordings were not from verified RA‐projecting LMAN neurons, and hence could have been interneurons or Area X projecting neurons. In light of this, we feel that modeling LMAN input to RA neurons as Poisson is justified. We have added this motivation to the text. To address the reviewers concern, however, we did probe whether having time‐varying LMAN inputs to RA impacts our conclusion. As the reviewer predicted, it does not. Strengthening and pruning of HVC‐RA synapses still decreases the variability in RA neuron firing (Figure 9), but because a strongly time‐varying LMAN input is less variable than a Poisson input, variability in RA, as compared to random LMAN input, is reduced. Again, this effect is largely independent of the effect on variability coming from reorganization of HVC‐RA synapses.

However, a single Poisson process may not adequately capture the spike train statistics of LMAN neurons, which tend to fire high‐frequency bursts amidst what can be well described as Poisson firing. Thus, we also extended our model of LMAN firing by adding Poisson‐like bursting. This also did not change the conclusion (see Figure 9), though made RA firing more variable, suggesting that bursting in LMAN is an efficient way of driving variability in the motor pathway, an interesting result in its own right.Author response image 3.Strengthening and pruning of HVC input to RA causes a reduction in variability regardless of whether LMAN firing is Poisson, Poisson with added Poisson bursts, or time varying. A. Average pair‐wise cross‐correlation (CC) between spike trains during different ‘song’ renditions in our model network assuming LMAN input with different structure and statistics. B. LMAN spike trains for 100 ‘song renditions’ simulated using (top) a Poisson process of constant rate, (middle) a Poisson process with added Poisson bursting, and (bottom) a time‐varying Poisson process. Blue curve shows the time‐varying modulation in the firing rate of the Poisson process.

*Also, it is surprising that the authors do not parameterize their model with the distribution of single-fiber synaptic strengths presented in the first part of the paper (the fact that model inputs are expressed in units of mV makes it hard to assess whether the model distribution is similar to the distribution observed experimentally). Would the key model results hold true using a more realistic distribution of input strengths*?

This is a great point, and we have now done what the reviewer suggested. We parameterize the strengthening and pruning to fit the experimental data and find that the key results from the simpler simulation still very much hold (Figure 8). We thank the reviewer for this suggestion as it makes our point even stronger.

Reviewer #3:

*Regarding novelty: prior anatomical and physiological studies have documented changes in the relative strength and numbers of HVC synapses on RA neurons over song learning in zebra finches, and also had characterized the intrinsic properties of RA neurons over a similar time course*.

We respectfully disagree on this point. We believe this misunderstanding stems from our manuscript not adequately distinguishing the current study from prior work. Below we draw this contrast and show that our results on how the number and strength of HVC input to RA changes during development could not have been inferred from prior studies.

We have added a paragraph to the manuscript contrasting our experiments with previous studies (though less detailed than the discussion below), to clarify this issue.

Relation of prior studies to our work:

One anatomical study from [25], shows that the number of synaptic inputs onto RA dendrites changes with development. Counting synaptic contacts in bulk RA tissue, however, does not say anything about the number of inputs to single RA neurons, or about the functional strength, or about number of HVC fibers that connect to single RA neurons.

Another paper specifically mentioned in the referee report was [33] study. The authors looked at various metrics of how the RA network reorganizes following lesions to LMAN and during normal development. The experiments reported on, however, were not designed to probe the logic by which HVC inputs innervate RA or how this changes with development.

One experiment we believe the reviewer may be referring to involved filling RA projection neurons and counting spines. Though spine numbers differ between juvenile and adult birds, there is no way to infer what this means in terms of changes to the number or strength of HVC inputs to single RA neurons. First, spines on RA dendrites reflect input from both LMAN and RA, and thus a change in spine number can’t be cleanly attributed to changes in HVC (or LMAN) input.

And even if they could somehow be parsed, synaptic inputs from HVC (and LMAN) are onto both dendritic shafts and spines, meaning that spines only tell half the story. Moreover, it is not clear how to relate the number of spines on RA dendrites to the number of unique HVC (or LMAN) inputs. Lastly, counting spines does not say anything about the functional strength of the inputs.

Another experiment in [33] may, at first glance at least, look similar to ours. The authors make sharp intracellular recordings in RA projection neurons and measure EPSPs evoked by HVC fiber tract stimulation. They show that the relationship between stimulation intensity and initial slope of the evoked EPSP changes.

However, the extent to which this effect is due to changes in the excitability of HVC fibers, faster polysynaptic response to HVC stimulation, more numerous HVC inputs or increased synaptic strength of HVC‐RA synapses simply cannot be parsed, and the authors state as much in their paper. Thus it is simply not possible to infer the distribution of synaptic strengths and relative number of inputs from this paper.

This is not meant as a criticism of Kittelberger and Mooney’s study. Their intention was not to uncover the functional logic by which HVC neurons innervate RA projection neurons. We have repeatedly consulted with Richard Mooney about our experiments, and he has encouraged us to perform them, as he believes, as do we, that they fill an important gap in our understanding of how the song system learns and develops (personal communication).

While we disagree with the assertion that the relative change in number and strength of HVC input to RA has been characterized, we agree that the intrinsic properties of RA neurons have been previously examined, and we acknowledge as much in the paper. We nevertheless repeated these experiments for three reasons: (i) to have measurement of intrinsic properties for the same age‐groups where we characterize the connectivity strength (Kittelberger and Mooney did one measurement in juvenile and one in adult); (ii) to interrogate the intrinsic properties in a more physiological range (higher temperatures and at higher instantaneous firing rates); (iii) to look at intrinsic properties with whole cell recordings rather than sharps, thus better controlling for leak currents. Our results are similar to those of Kittelberger and Mooney. As they are not providing a novel result (though we maintain they could have), we debated whether or not to include them in the paper. We decided to do this for the reasons we initially did the experiments (see above) and because we believe it is good for the field to see that results can be robustly replicated across labs and with different recording techniques.

*Thus, the characterization of LMAN synapses on RA neurons over song learning is the novel experimental data set provided here; the modeling study is also novel. Whether these together provide enough of an advance and will appeal to a broader readership is hard for me to assess. As a songbird researcher, I find the results very interesting and they help to articulate how the consolidation of HVC synapses on RA neurons could act as a counterweight to suppress noise from LMAN inputs. I would be interested to hear what the other reviewers' thoughts are about this*.

We appreciate the kind words, and hope that the reviewer agrees that there is also great virtue in better elucidating the logic by which the descending motor control system (i.e. HVC‐RA) is wired up during learning, and that our data provide new insights in this regard.

*Regarding mechanism-testing: the adult songbird's capacity to re-engage vocal plasticity and variability mechanisms through deafening or other perturbations of auditory feedback could more directly test the idea that the relative strength of HVC synapses is the major regulator of LMAN's ability to influence song variability. Whether the authors are still in a place to execute any more slice experiments is unknown, but an additional manipulation of this sort would strengthen support for their conclusions*.

We appreciate the spirit of this comment, and agree that it would be great to manipulate HVC‐RA connectivity and/or song variability in an experimentally tractable way, and directly relate the two. Deafening and auditory perturbations, however, are likely to affect and disrupt brain circuits well beyond the HVC‐RA circuit (Horita et al., 2008; Tschida and Mooney, 2012), making the interpretation of such experiments confounded and difficult. The current paper relates changes in RA circuitry during initial learning and development to the regulation of motor variability. It is not clear that increased variability after deafening or other perturbations would engage the same or similar mechanisms. Moreover, our contention is not that strengthening and pruning of HVC‐RA synapses is the only way to influence variability, and we now show that other mechanisms can contribute (Figures 8 and 9). Rather, the point we would like to make is that strengthening and pruning of HVC‐RA synapses is an important, and likely dominant, mechanisms for regulating variability during normal development.

[Editors’ note: what now follows is the decision letter after the authors submitted for further consideration.]

We thank the reviewers for their insightful comments in this and the previous round of review. The constructive comments have significantly improved the manuscript. Below we respond point-by-point to the reviewers’ comments.

Reviewer #2:

*In their revised submission the authors have significantly improved their manuscript by addressing some technical concerns regarding the analysis of their physiological data and by greatly expanding the scope and impact of their modeling work. In making these improvements (primarily the latter), the authors have convinced me. Although prior studies have investigated the issues they take on, the present manuscript represents a significant advance. Therefore, although it was not clear whether the original submission had sufficient impact for publication in* eLife*, I think that the revised manuscript represents a major contribution to the field and I think that it should be published. However there are a few things I would like to see addressed first*:

*1) The expanded modeling results greatly strengthen the authors' argument, however I think the presentation could be made a bit more clear. First, in*
Figure 6*, the accompanying legend, and the text in Results, I suggest that the authors more explicitly specify the baseline value relative to which the changes in LMAN input, FI gain, etc. are performed. For example, the legend to*
Figure 4
*reads: “…(c) strengthening/weakening LMAN inputs by up to 50%”. This should specify: “… up to 50% relative to the default value of W_LMAN, which was based on our empirical measurements of single-fiber currents (see Results;*
Figure 4*)”. Doesn't need to be this exact text of course, but for all parameters varied in*
Figure 6
*this type of information should be made more explicit. Also, if I'm reading*
Figure 6
*correctly, the black lines in 6C–E, G and H all represent the “default” model, this should also perhaps be made explicit in the legend. Finally, the lightest red dashed lines are nearly invisible in 6G and H. Perhaps these could be darkened to enhance readability*.

These are good points; we have now clarified, in the figure legend and text, which model we make the comparisons against. The lines in the figure have been darkened to make them more legible.

*2) In the Discussion section the authors consider how elimination and strengthening of HVC inputs to RA could separately contribute to a reduction of sensitivity to LMAN drive. However, the modeling studies never dissociate these two variables (number vs strength of HVC inputs), although it seems that the authors are in a good position to do so. This section of the Discussion is therefore somewhat unsatisfying and highlights that the authors have not performed an important variation on their analysis that could yield additional insight. I think that the authors should either explore this directly using the model (by dissociating input sparsity from input strength to back up their qualitative assertions) or remove this consideration from the Discussion*.

This is a great point and we have now done what the reviewer suggested. It turns out that pruning and strengthening connections from HVC to RA contribute to reducing variability on their own, but not nearly as much as when the two processes co-occur. I.e. the process of strengthening AND pruning has a much larger effect on variability than either strengthening OR pruning alone. We have added this result to Figure 6, and discuss the implications of this in the text.

Reviewer #3:

*I thoroughly enjoyed reading the greatly improved manuscript and the authors' response to the reviewers' comments, which largely addressed my concerns. This is an important study that goes to significant lengths to describe a developmental change in functional connectivity that could account for how motor variability decreases with skill learning. Overall, I believe that the study will appeal to a wide audience and serve as an extremely useful resource for those interested in neural mechanisms of motor variability and learning*.

*1) I was left to wonder whether the NMDA EPSC time course (i.e., decay kinetics) measured for LMAN inputs onto RA neurons are able to contribute to variability in RA neuronal firing in response to drive from HVC? The decay kinetics of NMDA-mediated currents at the LMAN to RA synapse are known to become faster with age, similar to developmental changes described at other vertebrate central synapses, especially in the mammalian neocortex. Although these changes are most often associated with sensory learning of various kinds, it would be useful to know whether a slower time course in the LMAN to RA synaptic current has an effect on RA spike variability*.

It’s a good question. We initially chose not to explore variation in NMDA kinetics, since available data from zebra finches suggest that it does not change over the course of sensorimotor learning (66). In response to the reviewer’s comment, however, we changed NMDA kinetics in our model. We find that even a fourfold change in the decay constant has only a minor effect on variability (see Figure 10 below). However, the simulations suggest that a decrease in this time constant, i.e. the trend seen during early development of neocortical NMDA synapses, actually increases variability, thus offering no explanation for the decreased variability in motor output. This result, however, may be confounded by another issue, which is that changing the time constant results in significant changes to RA firing rates. Decreasing the NMDA time constant from 100 ms to 50 ms, for example, reduces the average firing rate of model RA neurons from ∼50 Hz to ∼20 Hz. To keep the firing rate constant, we would need to tweak the model in ways that could also affect variability. Moreover, since most parameters in the model are already determined by our data, additional ‘tweaks’ would need to be justified. We note that changes to other parameters in the model over the ranges reported in the paper (Figure 6) did not materially affect the average firing rate of RA neurons.

Given that there is (i) no indication of a change in NMDA kinetics during sensorimotor learning in zebra finches, (ii) that changing the time constant in the model effects RA firing rates in a way that may confound the interpretations of the results, and (iii) since the effect on variability is modest, we would rather not include or discuss this result in the paper. Doing so, in a responsible way, would require quite a bit of elaboration, something we feel would take the focus away from the other results.Author response image 4.The effect of varying decay kinetics of NMDA currents in our model results in modest changes to RA variability, with longer time constants producing more stereotyped firing*.*

*2) I appreciate the authors' clarification of the novelty of their findings. I certainly agree that theirs “… is the first study to describe the functional changes in the inputs to RA from both HVC and LMAN during song development…”, and for that scope I believe the study is important and not merely incremental. And the detailed explanation they provided for why they view their characterization of the developmental changes at the HVC to RA synapse was also helpful, although I still maintain that those changes could be at least partly inferred from earlier structural and functional studies by Hermann, Kittelberger and Canady. This is not to minimize the current study's importance, but I do believe the authors could go further in integrating the earlier work into their present writing to better highlight exactly what is new here and also to more accurately place their study in the published literature*.

*With due respect to the authors' different perspective, I see pretty clear evidence of functional strengthening and consolidation at the HVC to RA synapse from those earlier studies when they are viewed collectively: over development, HVC bouton number in RA declines, spine density of RA projection neurons describes an inverted “U,” and the functional efficacy of the HVC to RA synapse increases. Further, Canady's serial EM reconstruction work (albeit in canary) revealed that HVC synapses are mostly (>90%) onto spines, whereas LMAN synapses are split between spines and shafts; because HVC synapses far outnumber LMAN synapses on single RA neurons (I believe the ratio was greater than 20:1, but it's been awhile since I have read it thoroughly), Canady concluded that the vast majority of inputs onto spines of RA dendrites reflected input from HVC and from other RA neurons and not from LMAN. I think it is reasonable to assume from this that the developmental changes in RA spine density reported by Kittelberger mostly reflect reorganization from those two sources and not from LMAN. This postsynaptic change is paralleled by a reduction in the frequency of putative HVC axonal boutons in RA; taken together, this seems to provide pretty clear structural evidence of a reduction in the number of HVC to RA synaptic connections with development*.

There seems to be a species-difference here as [25] report that, in zebra fiches, ∼40 % of HVC synapses are onto dendritic shafts. In zebra finches they also see a more balanced mix of synaptic contacts from HVC and LMAN onto RA dendrites than does Canady in canaries. That being said, we fully agree that the anatomical studies are consistent with our results and we now emphasize this further in the text. Yet there is no unambiguous way to predict from these structural studies how the organization of single fiber inputs to RA changes with development. This is because changes in the number of synaptic contacts could reflect changes in the number of functional inputs or fewer contacts per input. Moreover, the structural studies say little about the functional strength of the synapses. We believe our study, which focuses on the changes in the strength and number of single fiber inputs, nicely compliments previous anatomical work, giving us a more complete picture of the changes that accompany song learning.

*Lastly, those structural changes are accompanied by increased efficacy in the HVC to RA synapse, as measured by the stimulus current, EPSP onset slope relationship, which steepens significantly between 45 d juveniles and adults. At face value, this suggests to me that the HVC to RA synapse undergoes developmental consolidation in a process that involves synaptic pruning and strengthening*.

*Regarding determination of locus: Kittelberger's measurements of onset slope largely avoided the potential confound of polysynaptic components. He also found that spontaneous EPSC amplitudes increased (with LMAN lesions, which mimicked the developmental effects), making it less likely that the change in the stimulus-response relationship simply reflected changes in HVC fiber excitability. He noted a reduction in HVC bouton frequency along with decreased spine density and increased stimulus-response relationship. In aggregate, these findings strongly suggest to me a developmental process in which HVC synapses are pruned and strengthened, thus partially anticipating one aspect of the current study. Nonetheless, the current study goes well beyond this by providing more direct evidence of such consolidation while also providing an exact estimate of the numbers of inputs at a greater range of developmental points*.

We agree that Kittelberger’s study is consistent with the picture we find, but maintain that it is also consistent with other possibilities (continued developmental strengthening of single fiber inputs from HVC without pruning with each single fiber having fewer synaptic contacts, for example). We now acknowledge that the results from the Kittelberger study is consistent with ours, but point out that our findings provide essential new insight into the logic of how RA connectivity changes during learning.